# A scoping review of digital workplace wellness interventions in low- and middle-income countries

Yi Chiann Thai[1], Deanna Sim[2], Tracy A. McCaffrey[2]*, Amutha Ramadas[1], Hema Malini[3], Jessica L. Watterson[1]*

**1** Jeffrey Cheah School of Medicine and Health Sciences, Monash University Malaysia, Selangor, Malaysia, **2** Department of Nutrition, Dietetics and Food, Monash University, Melbourne, Australia, **3** Faculty of Nursing, Universitas Andalas, West Sumatra, Indonesia

* jessica.watterson@monash.edu (JLW); Tracy.McCaffrey@monash.edu (TAM)

**Data Availability Statement:** All data is available in existing library databases that we searched. Our methodology is outlined in the article to allow others to reproduce the search.

## Abstract

### Introduction

Digital technology-based interventions have gained popularity over the last two decades, due to the ease with which they are scalable and low in implementation cost. Multicomponent health promotion programmes, with significant digital components, are increasingly being deployed in the workplace to assess and promote employees' health behaviours and reduce risk of chronic diseases. However, little is known about workplace digital health interventions in low- and middle- income countries (LMICs).

### Methods

Various combinations of keywords related to "digital health", "intervention", "workplace" and "developing country" were applied in Ovid MEDLINE, EMBASE, CINAHL Plus, PsycINFO, Scopus and Cochrane Library for peer-reviewed articles in English language. Manual searches were performed to supplement the database search. The screening process was conducted in two phases and a narrative synthesis to summarise the data. The review protocol was written prior to undertaking the review (OSF Registry:10.17605/OSF.IO/QPR9J).

### Results

The search strategy identified 10,298 publications, of which 24 were included. Included studies employed the following study designs: randomized-controlled trials (RCTs) (n = 12), quasi-experimental (n = 4), pilot studies (n = 4), pre-post studies (n = 2) and cohort studies (n = 2). Most of the studies reported positive feedback of the use of digital wellness interventions in workplace settings.

### Conclusions

This review is the first to map and describe the impact of digital wellness interventions in the workplace in LMICs. Only a small number of studies met the inclusion criteria. Modest evidence was found that digital workplace wellness interventions were feasible, cost-effective,

**Funding:** The author(s) received no specific funding for this work.

**Competing interests:** The authors have declared that no competing interests exist.

and acceptable. However, long-term, and consistent effects were not found, and further studies are needed to provide more evidence. This scoping review identified multiple digital health interventions in LMIC workplace settings and highlighted a few important research gaps.

## Introduction

According to the World Health Organisation (WHO), 3.5 billion people, nearly half of the world's population, are employees. On average, a full-time employee spends more than one-third of his or her days, five days a week at their workplace. Due to the large population and long hours spent at work, the workplace has been a favourable setting to implement health promotion programs, motivating employee health behaviour change. WHO has estimated that 2.1% of all deaths and 2.7% of the global disease burden are attributed to quantified occupation risks [1]. Of these, employees in low-and middle-income countries (LMICs) have contributed to the largest portions of deaths and disability in the workplace settings. Evidence has suggested a rising need for WWPs in LMICs [2].

WWPs are typically designed to reduce medical spending, increase employee's productivity and enhance their well-being [3]. Research explored the link between employee health and work productivity [4, 5]. Besides absenteeism as an indicator for work productivity, there is also an extent of limitation due to health problems even when employees are present at work. For example, obese workers may experience greater challenges at work compared to normal weight workers [6]. Both absenteeism and presenteeism are strongly associated with poor employee's status and behaviours, including obesity, insomnia, depression or physical inactive which have been proven to cause detrimental burden to organizations' economic [7–11]. This poor workplace performance which is caused by physical or mental health issues is often underestimated by the organisation. Thus, it is essential to build a healthy work environment and address employee's health issues.

Studies have grown exponentially in a short time as digital health is progressing rapidly due to advances in technology and applications. Multicomponent design which involves various support from healthcare professionals, employees support groups, telephone-based coaching and more recently web and mobile-delivered programs, has been proven to be the most effective approach in addressing occupational health issues [12–17]. Digital technology-based intervention is increasingly being deployed in the workplace due in part to their scalability to a large population and cost-effective approach when compared with traditional health intervention used. Additionally, the COVID-19 pandemic has accelerated the digital transformation and brought more people on the digital health journey. The remote work model might affect the implementation of WWPs and thus, digital intervention may be more feasible and practical in this new norm. Also, digital workplace wellness allows all employees to access the health promotion content from anywhere at any time with the help of technology. Nonetheless, it is worth discovering whether digital workplace wellness is effective in modifying health behaviours. Employee populations potentially have much to gain from digital intervention for health behaviours promotion, yet little is known about the implementation of digital-based technology intervention in the LMICs workplace context as most of the studies reported were in developed countries. Hence, this scoping review aims to explore and provide a comprehensive synthesis of current evidence in relation to the effectiveness, feasibility, and acceptability of the digital workplace wellness intervention in the LMICs settings.

## Methods

### Study design

This scoping review utilises systematic searching methods and is guided by the Joanna Briggs Institute methodology [18] and Preferred Reporting Items for Systematic reviews and Meta-Analyses extension for Scoping Reviews (PRISMA-Scr) [19]. Ethical approval was not sought as the data were publicly available. A scoping review approach was chosen to examine the volume of existing literature and provide an overview of its focus, such as the digital components used, and health outcomes targeted. Specifically, we aim to explore the following research questions:

1. How have digital technology interventions been conducted in the workplace in LMICs?

2. What research has been done and what are the effects of these interventions on health- and job performance-related outcomes?

3. What are the research gaps that can be identified from these interventions as to improve health behaviours in the workplace in LMICs?

The protocol of this scoping review has been registered with Open Science Framework (10.17605/OSF.IO/QPR9J).

### Data sources and search strategy

We systematically searched 6 databases, including OVID Medline, Embase, PsycINFO, Scopus, CINAHL Plus and Cochrane Library. Four themes of keywords "digital health", "intervention", "workplace" and "developing country" were used to guide the search and a more detailed search strategy with relevant synonyms and medical subject heading (MeSH) terms, combined with Boolean Operator **AND** is provided below:

1. "Digital health" OR ehealth OR mhealth OR "mobile health" OR digital* OR web* OR internet* OR online OR smartphone* OR "cell phone*" OR "mobile phone*" OR telephone* OR application* OR "activity monitor" OR tracker OR pedometer OR technolog* OR messaging OR whatsapp* OR "whatsapp-based" OR "wechat*" OR "wechat-based" OR "social media"

2. Wellness OR wellbeing OR lifestyle* OR "workplace wellness" OR "occupational health" OR "health promotion" OR "health behavio?r*" OR intervention* OR program* OR education OR "physical activity" OR exercise* OR health* OR diet OR nutrition OR food OR "healthy eating" OR "mental health" OR "chronic disease*" OR sedentary OR "sedentary behavio?r*" OR stress* OR sleep*

3. Workplace OR employee* OR occupation* OR work* OR industr* OR office OR job* OR "job performance" OR "work engagement"

4. LMICs country list modified based on 2022 fiscal year classification [2].

The database search was limited to peer-reviewed original articles in the English language, human studies, and age group >18 years where possible restriction was imposed. Articles were only included from 2010–2021 to avoid the inclusion of obsolete digital components such as CD-ROMs and personal digital assistants (PDAs) which are not applicable in the current digital era. A sample of search strategy as performed in OVID MEDLINE on 15 December 2021 is shown in S2 Table.

This search strategy was refined after initial searches were run in October 2021 and resulted in few relevant studies. At the same time, the research team performed a manual search in

Google Scholar and identified relevant studies that were not included in the database results. The search terms were reviewed and revised, drawing on terms used in the publications identified manually, and with input from Monash University librarians. These search term revisions resulted in a higher number of relevant results, including the papers that had been located through manual searching, indicating they were more effective than the initial search terms. All searches were rerun with these search terms in December 2021.

## Study selection

Covidence software [20] was used to facilitate the screening process. All records retrieved were imported into Covidence where the duplicate records were automatically removed. Two reviewers first screened independently the titles and abstracts of the publications, and then proceeded to screen full text articles to determine the eligibility of the papers. Both reviewers screened all the articles. The screening was conducted according to predetermined eligibility criteria (Table 1). All conflicts were resolved through discussion with the research team. Emails were sent to corresponding authors as needed to get relevant information such as full texts of papers which could not be found online to further confirm the eligibility of the paper.

## Data extraction

Two reviewers conducted the data extraction from half of the 24 finalised articles (12 each). Information that was extracted included: country, year of publication, study design, participant characteristics, inclusion and exclusion criteria, intervention duration and follow-up, intervention components, measured outcomes, and main findings. As the scoping review is qualitative in nature, we performed narrative data synthesis according to five groups of identified outcomes (as shown in Table 3), namely Lifestyle (A) including smoking and cardiovascular disease risk, Weight Management (B), Physical Activity (PA) (C), Job performance (D) including work engagement, as well as other health outcomes (E) such as sleep, stress, and ergonomic condition. The reviewers worked independently on extracting the data of 12 studies each using a shared Google document, then later finalised and refined the data extraction table together. All conflicts were resolved through discussion with the research team.

**Table 1. Inclusion and exclusion criteria.**

| Inclusion criteria | Exclusion criteria |
|---|---|
| • Peer-reviewed publication<br>• Published in English language<br>• Conducted among adult employees in any work environment (excluding students alone) in any of the LMICs, as defined by the World Bank List of Economies 2022 [21]. This includes interventions in companies spanning multiple countries as long as at least one country is defined as an LMIC.<br>• Reported an intervention with at least one digital component (web/ mobile/ SMS/ phone applications)<br>• Presented qualitative or quantitative data relevant to health-related outcomes, work-related outcomes, or design-related outcomes (e.g., satisfaction, feasibility, acceptability, engagement, facilitators/barriers, perceptions about digital interventions).<br>• Any study design including pilot studies<br>• Wellness activities implemented by any parties (employer, government, third parties, etc.) | • Not conducted in human population<br>• Review articles<br>• Not a full-text, peer-reviewed article (book chapter, conference abstract, monograph, etc.)<br>• Protocols and study designs<br>• Did not report the details of the intervention and/or outcomes<br>• Special occupational groups without typical freedoms (e.g., soldiers), or with informal employment (e.g., sex workers) |

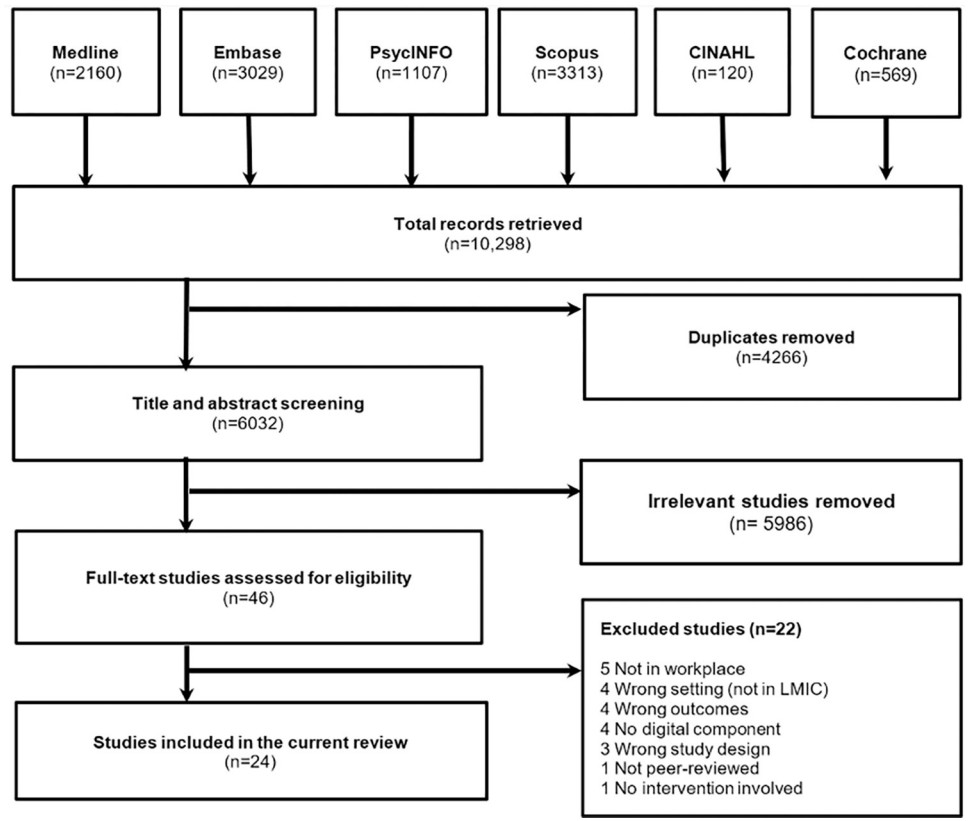

**Fig 1. PRISMA flow chart of study selection process including reasons for excluded studies.**

## Results

### Study selection and characteristics

Our database search identified a total of 10,298 studies. Fig 1 illustrates the PRISMA flow chart of the article selection process. After removing the 4266 duplicates from the imported studies and screening 6032 studies at the title or abstract level, we found 46 full text articles which were potentially relevant. Eventually, 24 studies were included in the current review after applying the inclusion and exclusion criteria.

A summary of the main characteristics of the included studies is provided in Table 2. All the studies were published between the years 2012 and 2021 (Fig 2). The study designs included randomized-controlled trials (RCTs) (n = 12), quasi-experimental (n = 4), pilot studies (n = 4), pre-post studies (n = 2) and cohort studies (n = 2). Six studies were conducted in China [22–27], four in Iran [28–31], four in India [32–35], two in Turkey [36, 37], two in Latin America [38, 39], one in South Africa [40], one in Brazil [41], one in Nigeria [42], and one in Vietnam [43]. Two studies were conducted in multiple countries [44, 45]: Ganesan A et al (2016) was conducted in LMICs in Asia and 90% of the total participants were from India and Montagni et al (2019) involved participants from China, France, Spain and the UK. For these studies including participants from both LMICs and high-income countries, results could not be separated by country, so the pooled results are presented in this review. Workplace settings included academic institutions [36, 40–42], hospitals or academic hospitals [22, 27, 31, 37, 39, 43], healthcare facilities [29, 30, 35], IT companies [25, 34], industrial units [33], and service companies [32]. Seven studies targeted public and private sector organisations from multiple worksites [23, 24, 26, 28, 38, 44, 45].

**Table 2. Tabulated results of included studies.**

| | | Total N |
|---|---|---|
| Country | Asia (China, Iran, India, Turkey, Vietnam) | 17 |
| | South America (Bolivia, Guatemala, Paraguay, Argentina, Guatemala, Peru, Brazil) | 3 |
| | Africa (South Africa, Nigeria) | 2 |
| | Multiple continents | 2 |
| Study Design | RCT | 12 |
| | Quasi-experiment | 4 |
| | Pilot | 4 |
| | Pre-post | 2 |
| | Cohort | 2 |
| Study Setting | Mixed | 5 |
| | Hospital and healthcare | 9 |
| | Industrial and Manufacturing | 1 |
| | Office | 5 |
| | University | 4 |
| Study Size | <100 | 9 |
| | 100–1000 | 12 |
| | >1000 | 3 |
| Population target | Normal | 17 |
| | At risk group (Overweight and obese, hypertension, stress etc.) | 7 |
| Gender | All | 22 |
| | Male-only | 1 |
| | Female-only | 1 |
| Underlying theory | Transtheoretical Model of Behaviour Change | 3 |
| | Social Cognitive Theory | 2 |
| | Health Belief Model | 2 |
| | Other theories | 5 |
| | Not reported | 11 |
| Goal/ Behaviour Targeted | Lifestyle / Chronic disease risk | 7 |
| | Weight Management | 6 |
| | Physical Activity | 4 |
| | Job Performance | 3 |
| | Stress and sleep | 5 |
| Intervention length | <3 months | 6 |
| | 3–12 months | 14 |
| | >12 months | 1 |
| Primary Outcome | Statistically significant improvement | 22 |
| | Non-significant | 2 |

## Theoretical frameworks

Some of the included studies drew on theoretical frameworks in their design or analysis. Three studies reported that their interventions drew on the Transtheoretical Model of Behaviour Change (n = 3) [33, 38, 40], with two [38, 40] used motivational messages and calls, and the other [33] providing health information. Two studies were based on Social Cognitive Theory (n = 2), with one [28] involving education/training, and the other [26] used a WeChat group

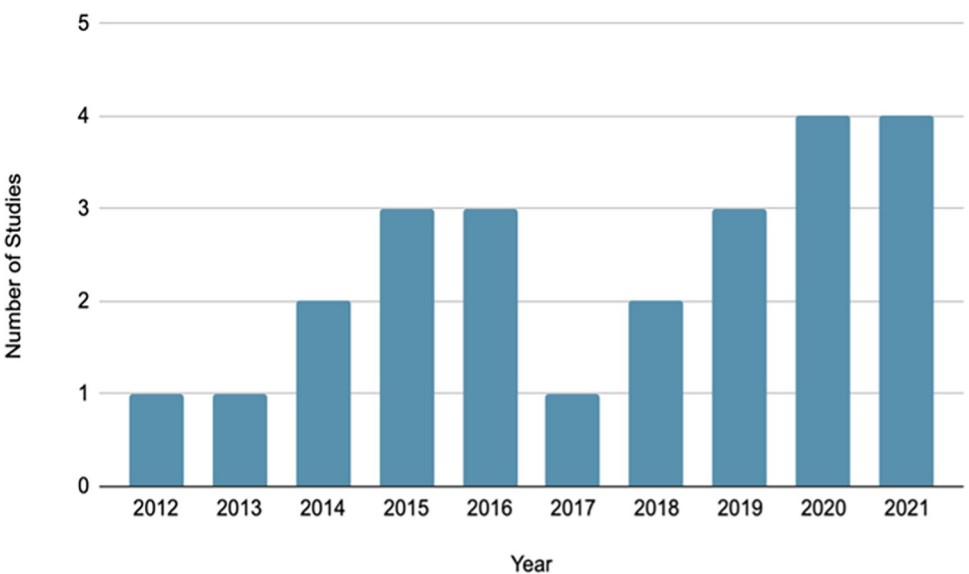

**Fig 2. Distribution of articles by year of publication.**

for motivation and progress reporting. Two used the Health Belief Model (n = 2), with one [30] involving education material and the other [38] involving motivational calls and personal text messages. One study adopted the Theory of Planned Behaviour [29] with education training, messaging and knowledge sharing in a Telegram group. Another used Behaviour Change Techniques [25] with coaching and pedometer-generated personalised feedback. Another used Self Efficacy [27] while asking participants to post 3 good things every day. One study used Goal Setting [34] and provided health information through phone messages and emails. Another study drew on the Behaviour Change Wheel [41] and involved coaching and pedometer-generated personalised feedback. Another study used Influential Theory [32] and involved pictures, videos, and text messages on positive emotions. Eleven studies mentioned no clear theoretical basis [22, 24, 31, 35, 37, 39, 43–45].

## Participant characteristics

The study sample sizes ranged from 41 in an RCT [31] to 26,562 in a prospective cohort study [44]. Of the 24 studies, two studies targeted single-gender participants: all male employees in the industrial sector [33] and all female staff working at a university [36], while all other studies involved participants of both genders. Overall, the proportion of females was higher than males, with 15 of 24 studies having ≥50% female participants. Employees' participation in all studies was voluntary and neither incentives nor monetary compensation were specified in any study. Thus, the drop-out rate for one of the studies was as high as 47% from baseline participation [44]. Ganesan et al (2016) used a competition method among countries and that may have motivated participants to complete the intervention. Most studies employed inclusion criteria that required participants to be adult employees above the age of 18 years. The study by Beleigoli et al. (2020) included both staff and students at a university and, though studies of students did not meet the inclusion criteria for this review, the pooled results are presented here as it was not possible to separate the results for staff only. Table 3 shows the study characteristics and main findings of the included papers. Supplementary materials are available to provide more details on the studies included.

**Table 3. Summary of the included studies (n = 24) grouped by main targeted health outcome.**

| Author, Country, Publication Year | Study Details | Intervention details | Measured Outcomes | Main findings |
|---|---|---|---|---|
| **A. Lifestyle/ Chronic disease risk (n = 7)** | | | | |
| Liu Z et al [22]., China, 2015 | **Type:** Clustered RCT **Participant:** 589 staff of hospital health management centre **Company size:** Not provided | **Length:**12-month intervention **Goal/Focus:** Reduce overall CVD (cardiovascular disease) risk **Mode of Delivery** **Intervention Group (IG):** receive mobile phone-based lifestyle intervention including an individualised electronic prescription, follow-up 5 to 8 min phone calls and text messages targeting reducing CVD risk during the 12-month intervention. **Control Group (CG):** receive usual medical examination without follow-up calls and text messages. **No theory applied.** | **Primary outcome:** Change in 10-year CVD risk between baseline and follow-up at 12-months, Change in components of risk score **Secondary outcome:** Diastolic blood pressure (BP), triglycerides (TG), high-density lipoprotein (HDL), and low-density lipoprotein (LDL), fasting plasma glucose (FPG), waist hip ratio (WHR) | **Significant difference between baseline and 12-month in CG:** 1.Increased mean CVD risk, Systolic BP (p<0.001), DBP mean value (p<0.001) 2.Decreased HDL and LDL (p<0.05), overall Diastolic BP, FPG and WHR (p<0.001) **Significant different between baseline and 12-months in IG:** Decreased 10-year CVD risk (Systolic BP, TC, BMI) (p<0.05), DBP mean value (p<0.001), HDL and LDL (p<0.05) Mobile phone-based intervention may therefore be a potential solution for reducing CVD risk in China. |
| Jorvand R et al. [30], Iran, 2020 | **Type:** Quasi-experiment **Participants:** 114 healthcare workers employed in two cities **Company Size:** 262 healthcare workers from two cities assessed for eligibility | **Length:** 2-week intervention with 6-month follow-up **Goal/Focus:** Effect of Health Belief Model (HBM)-based education on exercise **Mode of delivery** **IG:** Workers from one network received a Telegram-based intervention and supervised exercises, receiving education packages every 2 weeks and exercise reminder messaging, discussion and interlocution and their own exercise pictures sharing, **CG:** workers from the other network received uncontrolled individual exercise and self-reported. **Underlying theory:** HBM | **Primary outcome:** 1. **HBM** (perceived susceptibility, severity, benefits, barriers, and self-efficacy) 2. **Exercise** (daily and weekly in minutes) **Secondary outcome:** blood biochemical markers | **Significant difference of HBM constructs mean score at pre and post intervention in IG:** 1.Increased perceived severity (p = 0.000), perceived benefits (p = 0.010) and self-efficacy (p = 0.024) 2.**Decreased perceived susceptibility (p = 0.018):** directly related to increased preventive behaviours (doing exercise for CVDs) **Significant improvements of exercise in IG:** Daily exercise at post intervention (p = 0.001) Weekly exercise at pre and post intervention (p = 0.001) Educational interventions based-on Telegram messenger using HBM can improve exercise level. |
| Ramachandran A et al. [33], India, 2013 | **Type:** Prospective RCT **Participant:** 537 male employees in public and private industrial units **Company size:** Not provided, 8741 for eligibility screening | **Length:** 24-month intervention with follow-up **Goal/Focus:** Effect of phone messages on lifestyle change (PA and diet) to reduce Type 2 diabetes (T2D) **Mode of delivery** **IG:** Frequent mobile phone messages about healthy lifestyle, (the benefits, cues to start, ways to avoid relapse and remain motivated in physical activity (PA) and healthy dietary habits) **CG:** standard lifestyle modification advice at baseline only **Underlying theory:** Transtheoretical Model of Behavioural Change | **Primary:** Incidence of T2D **Secondary:** body mass index (BMI), waist circumference, systolic and diastolic BP, lipid profile, total dietary energy intake, PA score, acceptability of mobile phone messaging assessed by questionnaire | 50 (18%) men in the IG developed T2D over the 2 years compared with 73 (27%) CG (absolute risk reduction 9%) The intervention reduced the incidence of T2D during the study (β −0.447). The number needed to treat to prevent one case of T2D was 11 (95% CI 6–55). Mobile phone messaging is an effective and acceptable method to deliver advice and support towards lifestyle modification to prevent T2D in men at high risk. |
| Nurgul K et al. [36], Turkey, 2015 | **Type:** Descriptive and quasi experimental study **Participant:** 30 Female staff from Sakarya University **Company Size:** Sakarya University (44 staff voluntary participation) | **Length:** 3-month intervention without follow-up **Goal/Focus:** Effect of web-based education on health knowledge and behaviour **Mode of delivery** **IG:** 3-month web-supported health training material in ppt or audio-visual (3 modules: nutrition and health, PA, damages of smoking and stress management), available 7 days a week and 24 hours a day. **No CG or theory applied.** | **Primary outcome:** 1.**Knowledge on health promotion:** assessed by multiple choice questionnaire (MCQ) **Individual's health behaviours:** assessed via 52-item health promotion lifestyle profile (HPLSP) with 6 sub-dimensions (Health Responsibility, PA, Nutrition, Self-actualization, Interpersonal support, stress management) | **Significant difference (p<0.05) between pre and post intervention** 1.HPLSP total points (p = 0.001) 2.HPLSP sub-scale total points (p = 0.001) 3.MCQ (p<0.001) Web-based health education had a positive effect on healthy lifestyle behaviours of women staff working at Sakarya University and on their knowledge of health protection. |

*(Continued)*

**Table 3.** (Continued)

| Author, Country, Publication Year | Study Details | Intervention details | Measured Outcomes | Main findings |
|---|---|---|---|---|
| Rubinstein A et al. [38], Argentina, Guatemala, and Peru, 2016 | **Type:** Parallel RCT **Participant:** 637 adults with prehypertension from workplaces, health care, and community centres **Company size:** Not provided, 2630 for eligibility screening | **Length:** 12-month intervention with 6-month follow-up **Goal/Focus:** Effect of mHealth on cardiometabolic profile in prehypertension people **Mode of delivery** **IG:** Received monthly motivational counselling calls and weekly personalised messages about diet quality and PA. **CG:** Received usual care **Underlying theory:** Transtheoretical Model of Behavioural Change and the Health Belief Model | **Primary outcome:** Change in Systolic BP (mmHg) and Diastolic BP (mmHg) **Secondary outcome:** Change in weight (kg), BMI (kg/m2), WC (cm), PA (metabolic equivalent of task (METs)/ min per week), daily intake of fruits and vegetables (F&V), high fat and sugar foods | **Significant changes in highly engaged participants (received ≥75% of counselling calls in IG):** Body weight (-4·85kg), WC (-3·31cm), Daily intake of F&V (+0.66), Daily intake of high-sodium foods (-0.42), Daily intake of high fats food (-1.52) **Significant difference between IG and CG by country:** **1.Body weight in Peru (IG):** -1.24kg **2.Daily F&V intake in Peru (IG):** +0.64 **3.Daily F&V intake in Guatemala (IG):** +0.04 The mHealth-based intervention was associated with a small reduction in bodyweight and some dietary habits. A dose-response effect signalling potential opportunities for larger effects from similar interventions in low-resource settings was seen. |
| Martinez C et al. [39], Bolivia, Guatemala and Paraguay, 2018 | **Type:** Pre-post study **Participant:** 202 hospital workers from three organisations **Company size:** 1450 workers in three organization | **Length:** 6-month intervention with follow-up **Goal/Focus:** Effect of online training in smoking behaviour **Mode of delivery** **IG:** received an adapted version of a 5A's (Ask, Advise, Assess, Assist and Arrange follow-up) training program developed by the online platform e-oncología based on in-person courses. **No CG or theory applied** | **Primary outcome:** Cognitive and behavioural factors relating to smoking **Secondary outcome:** Self-reported performance level according to 5A's, demographics characteristic and questions suggested by experts to explore behavioural factors | **Significant Increase in performance of each of the 5A components (p<0.001)** **Significant improvement at post training** **1. Performance score of 5A's (p<0.001)** **2. Cognitive, behavioural, and organisational factors affecting 5A's** • **Five identified barriers (p<0.001):** self-reported preparedness, drug preparedness, competency in assisting smokers to quit, using additional resources, and having positive experience. • **Opportunities with score ≥7 (p<0.001):** motivation to help patients to quit, importance of smoking cessation in job, seeking frequently for patients. Online education on smoking cessation is feasible and effective in improving smoking cessation interventions in these countries. |
| Joseph-Shehu EM et al. [42], Nigeria, 2019 | **Type:** Pre-post study **Participant:** 22 university staff in Nigeria **Company Size:** 1349 staff | **Length:** 12-week intervention with follow-up **Goal/Focus:** Effect of information and communication technology on health promotion **Mode of delivery** **IG:** Adopted a nurse-client interactive Android phone app, Tertiary Staff Health Promotion App to access health promotion information, to monitor health status, increase PA, minimise sitting hours, with reminder of activities to improve health and quality of life. **CG:** N/A **Underlying theory:** Health promotion model | **Primary outcome:** Health promoting lifestyle behaviour Health status (BP, BMI, WHR, and fasting blood sugar) | **Significant difference between pre- and post-intervention:** Increased nutrition score (p = 0.0001), PA score (p = 0.0001), health responsibility score (p = 0.0001), stress management score (p = 0.001), Interpersonal relation subscales score (p = 0.009) Decreased BMI (p = 0.038) and diastolic BP (p = 0.04) The health-promoting lifestyle behaviour and health status of workers and other population groups showed improvements through information and communication technology. |

**B. Weight management (n = 5)**

*(Continued)*

**Table 3.** (Continued)

| Author, Country, Publication Year | Study Details | Intervention details | Measured Outcomes | Main findings |
|---|---|---|---|---|
| He C et al. [23], China, 2017 | **Type:** Cohort study<br>**Participant:** 15,310 employees from 134 government agencies and enterprises<br>**Company Size:** Not provided, 15818 for eligibility screening | **Length:** 6-month intervention with 6-month follow-up<br>**Goal/Focus:** Effect of WeChat-based multicomponent program on weight loss<br>**IG: Social media-promoted intervention.** Participants willing to use the research team's official WeChat account were enrolled in a WeChat group that provided feedback on weight, diet, and exercise weekly, microvideos and popular science knowledge on weight loss, community area for communication.<br>**CG:** Participants not willing to use official WeChat account given routine publicity such as slogan "take the stairs and lose weight" on weight loss<br>**No theory applied.** | **Primary outcome:**<br>**Weight loss:** height, weight, waist circumferences before and after intervention<br>**Demographic characteristics:** gender, age, educational level, and telephone number online registered with WeChat account | **Weight loss: IG** (2.09±3.43kg) > CG (1.78 ±2.96kg)<br>**Effect of WeChat on weight loss (Assessed with propensity method, p<0.05):**<br>1. Males in IG (active or inactive) had higher probability of maintaining weight with 1-2kg or > 2kg weight loss than CG (0-1kg)<br>2. Active participants in WeChat groups were more likely to lose weight.<br>The weight loss intervention campaign based on an official WeChat account focused on an occupation-based population in Shunyi District was more effective for males than females. |
| Yu Y et al. [24], China, 2018 | **Type:** RCT<br>**Participant:** 802 employees from institutions or enterprises from 4 areas, 44.9% overweight & obese, 49.6% normal weight and 0.5% underweight<br>**Company Size:** Not provided, 904 for eligibility screening | **Length:** 3-month intervention<br>**Goal/Focus:** Effect of pedometer-based walking and diet guidance on weight management<br>Mode of delivery<br>**IG:** Receive self-monitored intervention trial (exercise prescription and dietary guidance), synchronise pedometer exercise data to the Internet-based Health System Centre daily (at least weekly)<br>**No CG or theory applied** | **Primary:** Changes in body weight (kg) or BMI, waist circumference (cm), and BP (mmHg).<br>**Secondary:** Changes in lifestyle behaviour (scores), body fat percentage (%), fasting blood glucose/ fasting serum glucose (mmol/L), and serum lipid (mmol/L) | **Normal weight participants:**<br>1. Weight decreased 0.7% (p < 0.01).<br>2. Body fat percentage decreased 2.5% (p < 0.01).<br>3. BP and FSG decreased significantly (p<0.05)<br>**Underweight participants:**<br>1. Weight gain of 1.0% (p< 0.05),<br>**Overweight participants:**<br>1. 68.2% (208/305) experienced weight loss, with an average reduction of 3.5%, with 20.2% (42/ 208) of them achieving weight loss 5%.<br>2. BP and FSG decreased significantly (p<0.05)<br>The incidence of hypertension was significantly lower and lifestyle behaviour significantly improved (p < 0.05) |
| Abdi J et al. [28], Iran, 2015 | **Type:** Three-arm RCT<br>**Participant:** 435 governmental employees with overweight or obesity (BMI>25kgm/m2)<br>**Company Size:** Not provided,1200 for eligibility screening | **Length:** 6-month intervention with 3-month follow-up and maintenance<br>**Goal/Focus:** Effect of communication technologies and social-cognitive based education on weight loss<br>**Mode of delivery**<br>Two IG received lifestyle program and general brochures:<br>**IG1 (Web-assisted group):** educational content provided on a website.<br>**IG2 (Telephone-assisted group):** educational content provided through SMS every two weeks<br>**CG:** only receive general brochures about lifestyle and overweight<br>**Underlying theory:** Social cognitive theory (SCT) | **Primary:** anthropometric measures include weight (kg), waist circumference (cm) and blood pressure (mmHg)<br>**Secondary:** SCT measures | **The lifestyle intervention resulted in a weight loss of:** -1.08 kg in IG1, -1.92kg in IG2<br>**IG1:** mean scores of the constructs of self-efficacy (P = 0.001) and outcome expectancies (P = 0.020) increased.<br>**IG2:** mean scores of the constructs of self-efficacy (P = 0.001), environment (P = 0.001), outcome expectations (P = 0.040), and outcome expectancies (P = 0.001) increased.<br>A significant difference (P = 0.03) was observed in weight loss among the groups over the course of time. |

*(Continued)*

**Table 3.** (Continued)

| Author, Country, Publication Year | Study Details | Intervention details | Measured Outcomes | Main findings |
|---|---|---|---|---|
| Limaye T at al. [34], India, 2017 | **Type:** RCT<br>**Participant:** 265 young Indians of IT industry with normoglycemia and at high risk of developing diabetes.<br>**Company Size:** Not provided, 437 for eligibility screening | **Length:** 1-year intervention<br>**Goal/Focus:** Effect of technology-based lifestyle program on reducing T2D risk<br>**IG:** Receive lifestyle modification in Informative technology (LMIT) program<br>1. **Mobile phone messages and emails** with infographics which contain lifestyle modification information provided. No message was repeated.<br>2. **Additional support through a website and a** closed Facebook group.<br>**CG:** no intervention received<br>**Underlying theory:** goal setting | **Primary outcome:** prevalence of overweight/ obesity (BMI ≥ 25 kg/m2)<br>**Secondary outcomes:** Change in weight, waist circumference, blood pressure, glucose, lipid, lifestyle choices, diabetes awareness score, acceptability, and cost-effectiveness of the intervention | **Significant weight loss in IG (p<0.05):** Overweight/obese participants decreased from 104 (78.2%) to 96 (72.2%)<br>**Significant weight gain in CG (p<0.05):** Overweight/obese participants increased from 101 (76.5%) to 110 (83.3%)<br>The number needed to treat/prevent one case of overweight/obesity in 1 year was 9.<br>A virtual assistance-based lifestyle intervention was effective, cost-effective and acceptable in reducing risk factors for diabetes in young employees in the IT industry and is potentially scalable. |
| Beleigoli A et al. [41], Brazil, 2020 | **Type:** Three-arm parallel RCT<br>**Participant:** 1,298 overweight and obese students and staff of a Brazilian university<br>**Company Size:** Not provided, 3745 for eligibility screening | **Length:** 24-week intervention<br>**Goal/Focus:** Effect of personalized web-based coaching on weight loss with overweight and obese people<br>**Mode of delivery**<br>**Platform-only group (IG1):** received reminder emails to report weight and habits at 12 and 24 weeks after baseline trial and get access to 24-week web-based weight loss program with personalised computer-delivered feedback.<br>**Platform and coaching group (IG2):** received 24-week web-based weight loss program with 12-week personalised dietician-delivered feedback.<br>**CG (waitlist):** received a non-personalized minimal intervention based on dietary and PA recommendations delivered through downloadable e-booklet and four 5-min videos.<br>**Underlying theory:** Behaviour Change Wheel | **Primary outcome:** changes in weight and BMI<br>**Secondary outcome:**<br>1.**Changes in dietary intake:** assessed by the daily F&V portions, weekly consumption of sweetened beverages and ultra-processed foods<br>2. **Changes in PA:** Moderate and vigorous PA was assessed by the Brief PA Assessment Questionnaire. | **After 24-week intervention:**<br>1.Primary outcome (Weight & BMI change)<br>• **CG:** Weight = -0.66kg, BMI = -0.24kg<br>• **IG1:** Weight = -1.08kg, BMI = -0.38kg<br>• **IG2:** Weight = -1.57kg, BMI = -0.56kg<br>• **Significant overall weight loss (p = 0.001):** IG1 (83/420, 19.8%) and IG2 (64/408, 15.7%) > CG (61/270, 13.0%)<br>2.**Secondary outcome:**<br>• **Significant increase in F&V consumption (p = 0.001):** IG1 and IG2 > CG<br>• **Significant reduction in ultra-processed food consumption (p = 0.005):** IG1 and IG2 > CG<br>• **Significant increase in sweetened beverage consumption (p = 0.02):** IG1> IG2 |
| **C. Physical activity (PA) (n = 4)** | | | | |
| Blake H et al. [25], China, 2019 | **Type:** Two-arm clustered RCT<br>**Participant:** 282 employees of IT private sector organizations in Beijing and Gaungzhou<br>**Company size:** 690 employees from two sites | **Length:**12-week intervention without follow-up<br>**Goal/Focus:** Effect of digital video-based exercise on PA and work performance<br>**Mode of delivery**<br>**IG (Guangzhou):** "Move-It" digital video-based worksite exercise intervention<br>1. **Move-It website:** Six 10-min Qigong exercise video clips, twice/day on working day<br>2. **Reminder messages:** The Move-it desktop icon popped up at the same time, twice/day<br>3. **Exercise adherence info:** Daily exercise logs were collected through Move-it icon<br>**CG (Beijing, wait-list):** received the same intervention after the intervention period<br>**Underlying theory:** Behaviour Change Techniques (BCTs) | PA level, weekday sitting hours and work performance | **Significant differences at post intervention:**<br>1.**Increased PA:**<br>• IG (5.80 hrs/week, p = 0.04)<br>• CG (7.41 hrs/week, p = 0.00)<br>2. **Work performance:**<br>• Increased in CG (+0.69 units, p = 0.01)<br>• Reduced in IG (−0.03 units, p = 0.78)<br>3.**Increased sitting hours:**<br>I • G (+10.34 hrs/week, p = 0.00)<br>• CG (+5.68 hrs/week, p = 0.00)<br>• Between groups (−4.66 h/w, p<0.01)<br>The intervention did not result in greater changes in the IG than in CG. Many participants perceived the Qigong exercises positively and reported positive benefits on physical and mental health including muscle relaxation, stress reduction and improved working mood. |

*(Continued)*

**Table 3.** (Continued)

| Author, Country, Publication Year | Study Details | Intervention details | Measured Outcomes | Main findings |
|---|---|---|---|---|
| Gu M et al. [26], China, 2020 | **Type:** Quasi-experiment with self-controlled design<br>**Participant:** 262 workers from 17 worksites<br>**Company Size:** Not provided, 398 employees participated in study baseline | **Length:** 100-day intervention with follow-up<br>**Goal/Focus:** Effect of pedometer and WeChat-based group program on PA and health-related outcome<br>**Mode of delivery**<br>IG: a pedometer- and group-based intervention program with 10–20 participants each in 47 groups.<br>1.**Pedometer:** to monitor the PA during waking hours and upload the data to a specific website.<br>2.**We-Chat group:** created by each group captain to share daily steps number, communicate, and motivate the participants to achieve the corresponding goals.<br>CG: required to complete the measures and no intervention applied.<br>**Underlying theory:** SCT | **Primary outcome:**<br>1.PA<br>2.**Health related outcome:** height (cm), weight (kg), Body Fat % (BF%), systolic diastolic BP, waist and hip circumference, BMI<br>**Secondary outcome:** Job demand and control which measured based on Karasek's Job Content Questionnaire (JCQ) | **Baseline**<br>**Significant difference between IG and CG** (p<0.05): Waist and hip circumferences, BMI<br>**Post intervention**<br>• Walking increased about 22% compared with baseline vigorous PA<br>• Significant increase in vigorous PA (p = 0.048) and walking (p<0.01) in IG<br>• Significant reduction in moderate PA in both IG (p<0.01) and CG (p<0.01)<br>• Significant reduction in health-related outcomes at post intervention (p<0.05) in IG: systolic BP, waist and hip circumferences, body fat and BMI<br>**Significant association between:**<br>1.**Gender and WHR (p<0.001):** females showed larger decrease WHR<br>2.**Age and BF% (p = 0025):** age increased; BF% decreased<br>3.**Age (p<0.027) and difference in METs for vigorous PA (p<0.001) with BF%:** older and higher difference showed larger decreased BF%<br>4.**Vigorous PA and BMI (p = 0.013):** higher difference, larger decreased BMI |
| Pillay JD et al. [40], South Africa, 2014 | **Type:** Pilot study<br>**Participant:** 22 Staff members of the Faculty of Health Sciences, University of Cape Town<br>**Company Size:** Not provided, 25 employees agreed to join | **Length:** 10-week intervention with 2-week follow-up<br>**Goal/Focus:** Effect of pedometer-based program on PA level<br>**Mode of delivery**<br>IG: Received biweekly individual email feedback, motivational messages, and strategies to increase PA based on electronic receipt of pedometer data.<br>CG: Received general motivational message biweekly without pedometer feedback<br>**Underlying theory:** Theoretical model of behavioural change | **Primary outcome:**1.<br>**Participants' perceptions:** false, appeal, support and benefits of the intervention assessed through questionnaire during follow-up period.<br>**Secondary outcome:**<br>1.**PA level = measured at baseline and follow-up**<br>• Steps per day: recorded by pedometer Biometric and clinical measure, including waist circumferences (cm), body fat (%), BMI (kg/m2), systolic and diastolic blood pressure (mmHg) | **IG at 2-week follow up period after intervention:**<br>1.Average daily aerobic steps: decreased (-54 ±2746 steps)<br>2.Daily aerobic time (min): Increased (+0.9 ±23.0)<br>3.Daily steps: increased (+996±1748 steps)<br>**CG at 2-week follow up period after intervention:**<br>1.Daily steps: increased (+97±750 steps)<br>This pilot study provides useful information on the potential for PA improvements through pedometry in an employed, adult group. |
| Ganesan A et al. [44], Asia (India), Europe, Africa, North America, South America, Australia, and New Zealand, 2016 | **Type:** Prospective cohort study<br>**Participant:** 26,562 Indian employees from private and public sector organizations<br>**Company Size:** Not provided, 69,219 employees completed pre-event questionnaire | **Length:**100-day intervention annual program without follow-up<br>**Goal/Focus:** Effect of mHealth on PA, sitting and weight<br>**Mode of delivery**<br>IG: Stepathon workplace-based pedometer program<br>1.**Pedometer:** for step count challenge.<br>2.**Stepathlon website** (desktop and mobile version): provides personalized tools, educational content, and online community platform.<br>3.**Daily encouraging emails:** PA and nutrition messages, entertaining quizzes, and competitions to encourage online interface interaction<br>**No CG or theory applied.** | **PA measures:**<br>Change in step count, sitting duration and weight (kg) | **Post program showed significant improvements (p<0.0001) in:**<br>1.Recorded step count (+3,519 steps/day)<br>2.Exercise days (+0.89 days)<br>3.Sitting duration (-0.74 h)<br>4.Weight loss (-1.45 kg)<br>Improvements occurred in women and men, in all geographic regions, and in both high and lower-middle income countries. |

**D. Job Performance (n = 2)**

*(Continued)*

Table 3. (Continued)

| Author, Country, Publication Year | Study Details | Intervention details | Measured Outcomes | Main findings |
|---|---|---|---|---|
| Guo YF et al. [27], China, 2020, | **Type:** RCT<br>**Participant:** 73 clinical nurses from Chinese tertiary general hospital<br>**Company Size:** 197 nurses | **Length:** 6-month intervention without follow-up<br>**Goal/Focus:** Effect of WeChat-based psychotherapy on job performance and self-efficacy of people suffering burnout.<br>**Mode of delivery**<br>**IG:** WeChat-based 3GT-positive psychotherapy.<br>1.**WeChat:** to record three good things that were impressive each day and answer 2 questions: "Why did these good things happen?" and "What was your role in bringing them about?" in their WeChat circle<br>2.**Reminder messages for recording 3 good things:** sent to all the nurses at 8 pm by the researcher to remind them to increase the adherence of 3GT<br>**CG:** No intervention received<br>**Underlying theory:** Self Efficacy | **Primary outcome:** job performance measured by a 16-item scale<br>**Second outcome:** Self-efficacy was measured by the 10-item GSS<br>**Tertiary outcome:** burnout was measured by MBI-GS | **Significant main intervention effect** ($p<0.05$) **and interactions** ($p<0.05$) in both IG and CG on job performance and self-efficacy.<br>**Significant difference** ($p<0.05$) **between post-**intervention scores for job performance and self-efficacy between IG and CG.<br>**Significant difference** ($p<0.05$) **between scores** for job performance and self-efficacy of the IG before and after.<br>Three Good Things are recommended to be included into the management systems to improve nurses' physical and mental health and work outcomes over the long term. |
| Sasaki N et al. [43], Vietnam, 2021 | **Type:** Three-arm RCT<br>**Participant:** 951 nurses from national public tertiary hospital<br>**Company Size:** 1256 nurses | **Length:** 7-month intervention with follow-up<br>**Goal/Focus:** Effect of phone-based stress management program on work engagement<br>**Mode of delivery**<br>**IG:** Two 6-module CBT programs using ABC Stress Management app. **Program A** with free-choice multi-module, **Program B** with fixed-sequential order multi-module, both receive weekly messages and access to informal group chat (via social media apps such as Vober, Zalo, FB messenger) to receive intensive technical support<br>**CG (waitlist):**<br>1.Free to use any other mental health services as usual treatment during the intervention period.<br>2.Received intervention after the 7-month intervention period<br>**No theory applied.** | Work engagement scores | **Work engagement scores in both IG:** Increased from baseline to 3-month follow-up but decreased at the 7-month follow-up<br>**At 3-month follow-up:**<br>1.**Program A** showed a **non-significant** trend (P = 0.07) toward improved engagement.<br>2.**Program B** showed a **significant** intervention effect on improving work engagement (P = 0.049) with a small effect size.<br>**7-month follow up:** neither program achieved effectiveness.<br>A fixed order (program B) delivery of a smartphone-based stress management program improving work engagement in nurses in Vietnam effectively but with temporary and small effect. |
| **E. Other health outcome such as stress (n = 3), sleep (n = 2), ergonomic condition (n = 1)** | | | | |
| Aliakbari R et al. [29], Iran, 2020 | **Type:** Quasi-experimental study<br>**Participant:** 63 general dentists and dental specialists in Bojnourd city<br>**Company Size:** 90 general dentists | **Length:** 3-month intervention without follow-up<br>**Goal/Focus:** Effect of digital-based education to improve occupational health and ergonomic condition<br>**Mode of delivery**<br>**IG:** educational intervention developed based on predictive constructs using modern media.<br>**Messages:** Receive 1–2 per day for a month about changing behaviour, share knowledge in a Telegram group to improve musculoskeletal conditions, and receive access to online educational material<br>**CG:** received software package consisting of several applications, articles and corrective trainings as IG but not involved in Telegram group.<br>**Underlying theory:** Theory of Planned Behaviour | **Primary outcome:** evaluated by a questionnaire about<br>1.**Health condition:** personal info, daily activities, and exercise<br>2.**Knowledge and constructs of behavioural intention model**<br>**Ergonomics condition:** measured by Nordic questionnaire and Rapid Upper Limb Assessment (RULA) | **Significant difference of mean scores of constructs between IG and CG at pre-intervention:** Perceived control (p = 0.04)<br>**Significant difference of mean scores of constructs in IG at post-intervention:** Attitude score (p = 0.03)<br>Subjective norms, perceived control, attitude, and behavioural intention had the highest predictive power in improving the health and ergonomic position of dentists, respectively. |

*(Continued)*

Table 3. (Continued)

| Author, Country, Publication Year | Study Details | Intervention details | Measured Outcomes | Main findings |
|---|---|---|---|---|
| Nourian M et al. [31], Iran, 2021 | **Type:** RCT<br>**Participant:** 41 nurses working in 2 COVID-19 care wards in hospital.<br>**Company Size:** 44 nurses in 2 COVID-19 care wards | **Length:** 7-week intervention<br>**Goal/Focus:** Effect of WhatsApp-based training program on sleep quality<br>**Mode of delivery**<br>IG: Mindfulness-based stress reduction training program via WhatsApp group. Training content included educational media files of mediation, yoga exercise, speeches delivered by professionals<br>CG: completed 2 questionnaires on Porsline website, received music and training file without WhatsApp application.<br>**No theory applied.** | Sleep quality measured by Pittsburgh Sleep Quality Index tool (Score of 5 or higher indicates poor sleep quality) | **Significant improvement on sleep quality score in IG:** Subjective sleep quality (p<0.05), Sleep latency (p<0.05), Habitual sleep efficiency (p<0.05)<br>**Significant improvement on sleep quality score in CG:** Subjective sleep quality (p<0.05), Daytime drowsiness (p<0.05), Total sleep quality score (p = 0.001)<br>**Significant difference on sleep quality score between IG and CG:** Sleep latency (p<0.05), Subjective sleep quality (p<0.001)<br>The total sleep quality did not change among the participants in the IG at pre- and post-intervention but increased significantly in the CG. The MBSR program may be effective in improving the sleep quality of nurses. |
| Pendse et al. [32], India, 2012 | **Type:** Pilot study with part 1 (P1) survey and part 2 (P2) intervention<br>**Participant:** Service sector employees: 81 (P1) and 10 (P2)<br>**Company Size:** Not provided | **Length:** 2-week intervention without follow-up<br>**Goal/Focus:** Effect of web-based intervention on work life quality and mental well-being<br>**Mode of delivery**<br>IG: receive two 20–40 stimuli in the forms of pictures, videos, and text through official emails every day for 10 working days.<br>CG: No intervention received<br>**Underlying theory:** Influential theory | **Questionnaire in P1:** measure career and job satisfaction, perceived absence of work stress<br>**Questionnaire in P2:** measure affect balance and emotion/worry | **Happiness positively and significantly related to**<br>1. Quality of Work Life (R = 0.378, P<0.01)<br>2. Resilience (R = 0.365, P<0.05)<br>CG on affect balance significantly lower than the gain scores of IG (U = 14, P<0.05)<br>Web-based interventions show promise for enhancing employee's happiness but this study was limited by there's limitation on small sample size |
| Divya K et al. [35], India, 2021 | **Type:** Pilot study with single arm pre-post design<br>**Participant:** 92 healthcare providers (HCPs)<br>**Company Size:** 7597 HCPs from different states | **Length:** 40-day intervention with follow-up<br>**Goal/Focus:** Effect of online video-based education on mental wellbeing<br>**Mode of delivery**<br>IG: received a 4-day online breath and meditation workshop, Sudarshan Kriya Yoga (SKY) delivered by trained instructors with a 2-hour session/ day through video conference. Participants also learnt the 35-min home practice, including Pranayama, Bhastrika and SKY breathing to be practised at home daily.<br>**No CG or theory applied.** | **Primary outcome:**<br>1. Depression and Anxiety: measured by self-reported Depression, Anxiety and Stress Scale (DASS-21).<br>2. Sleep Quality: measured by Pittsburgh Sleep Quality Index (PSQI).<br>3. Resilience: measured by self-rated Connor-Davidson Resilience Scale.<br>Life satisfaction: measured by a 5-item Satisfaction with Life Scale. | **Significant differences in the scale scores:**<br>1. Depression, anxiety, and stress: reduced scores for all at post-intervention (p<0.001)<br>2. Resilience: Increased resilience at post-intervention (p<0.001) and greater increase at follow-up phase (p = 0.015).<br>3. Life satisfaction: increased life satisfaction at post-intervention (p<0.001) and greater increase at follow-up phase (p< 0.001).<br>4. Quality of sleep: reduced scores immediately after the program (p<0.001)<br>SKY breathing technique had a positive impact on the well-being of healthcare professionals during the pandemic. Participants experienced improved quality of sleep, enhanced satisfaction with life, and increased resilience after SKY. |
| Dincer B and Inangil D [37], Turkey, 2021 | **Type:** RCT<br>**Participant:** 72 nurses caring for COVID-19 patients in a university hospital<br>**Company Size:** Not provided, 80 nurses met criteria | **Length:** Not provided, without follow-up<br>**Goal/Focus:** Effect of online group emotional freedom techniques (EFT) treatment on reducing stress, anxiety, and burnout level<br>**Mode of delivery**<br>IG: received a 20-mins guided online group EFT treatment by showing the participants a picture of the acupressure points and ways to apply pressure tap.<br>CG: no EFT treatment received between the completion of 2 subjective units of distress (SUD) and burnout tests.<br>**No theory applied.** | **Primary outcome:**<br>1. Stress levels: measured by SUD scale<br>2. Anxiety levels: measure the State Anxiety Scale<br>3. Burnout levels: measured by a 21-item Burnout Scale | **Significant differences in IG at post intervention** (p < .001):<br>1. Stress levels: Reduced mean SUD score<br>2. Anxiety levels Reduced anxiety score<br>3. Burnout levels: Reduced burnout score<br>CG showed no statistically significant changes on these measures (p > .05) |

*(Continued)*

**Table 3.** (Continued)

| Author, Country, Publication Year | Study Details | Intervention details | Measured Outcomes | Main findings |
|---|---|---|---|---|
| Montagni I et al. [45], China, France, Spain, UK, 2019 | **Type:** Pilot study **Participant:** 291 employees (T1) of eight company sites in four countries **Company Size:** Not provided, 834 employees in (T0) | **Length:** Two phase (T0 and T1) intervention of 5 days each. Follow-up (T1) after 6 months **Goal/Focus:** Effect of blended tablet-based survey to raise sleep awareness **Mode of delivery** **IG:** asked to use WarmUapp tablet application which consist of 27 screens (23 screens for questions, 3 screens for partial survey answers, 1 screen for survey results and personalised recommendations to improve sleep quality) **No CG or theory applied** | **Primary outcome:** 1.Change in sleep status which measured total sleep duration, sleep efficiency, sleep debt, insomnia, sleepiness 2.**WarmUapp™ effectiveness:** measured user satisfaction, feedbacks and ideas through structured interview composed of 3 open-ended questions | **Significant difference in sleep status at post intervention (follow-up phase):** **1.Increased total sleep duration (p = 0.046)** **2.Decreased sleep debt (p = 0.019), sleep difficulties between two phases (p<0.001), sleepiness (p = 0.026)** **3.Sleep problem:** females 2 times more likely to suffer in both phases (p = 0.006) **Satisfaction with WarnUapp™:** All interviewees were satisfied of the intervention Interventions blending face-to-face and web-based approaches show promise for effective promotion of sleep awareness at the workplace. |

## Target population

17 studies targeted general employees without any health condition and seven targeted the at-risk population. Four articles employed a weight management intervention with an overweight or obese population (n = 3) [28, 41, 46] or population intending to lose weight (n = 1) [23]. One article included employees having prehypertension, one included employees who showed high stress symptoms, and one included employees with a family history of risk factors of metabolic diseases. Of the 24 articles, two targeted at male-only [33] and female-only [36] population, the remaining included both genders.

## Intervention and study characteristics

**Duration of intervention and follow-up.** The duration of intervention delivery varied, ranging from 2 weeks to 2 years, with six studies running from 1 to 10 weeks, six lasting for 3 months, five studies for 6 months, three studies for 12 months and two studies for 24 months. The two shortest interventions lasted for 10 working days. One [30] involved education and messaging through Telegram, the other [32] provided two 20–40 second stimuli in the form of pictures, videos and text through emails. 11 studies [23, 26, 28, 30, 33, 35, 38, 40, 43, 45] had follow-up periods after the intervention ranging from 2 weeks to 6 months. However, most of the studies did not include a detailed description of their follow-up phase, if any.

**Intervention provider.** The majority of studies involved interventions provided by the research team. Seven studies included interventions led by both the research team and participants. Of these, one study chose some participants to serve as role models for the remaining participants (using observational learning) while the research team were delivering educational content and consultations [28]. Two studies assigned some participants as leaders to facilitate the interventions and motivate participants to engage [25, 26]. Sasaki et al (2021) included participants in the program development, exploring the cultures and specific stressors in the workplace context. Additionally, Montagni and her team [45] built local teams in each participating country to lead adaptation and implementation of the pilot intervention with instructions from headquarters. For cohort studies which only involved data collection and analysis, one was a collaborative project with 2 universities (intervention provider) [44] and the other collaborated with the district government (intervention provider) [23].

**Digital component.** The digital components of interventions included: educational content or video demonstrations shared on websites or by SMS, email consultations, telephone counselling, use of a pedometer (piezoelectric accelerometer technology), WeChat and WhatsApp groups (communication mobile applications), and smartphone applications. 10 studies [23, 32, 33, 35, 39, 43, 45] involved a single digital component, whereas 14 studies [22, 24, 31, 34, 40, 41, 43, 44] utilised interventions where more than one digital component was used. For example, Yu Y. et al (2018) used an individualised pedometer-assisted exercise prescription and asked participants to synchronise the data to the Health System Centre daily. Abdi J. et al (2015) used websites and SMS to deliver healthy nutrition and PA information, as well as providing telephone and email consultations every two weeks. Of the 24 studies, 7 studies [24, 25, 33, 36, 39, 42, 45] involved only a digital intervention, while the remaining 16 studies [22, 23, 26, 28, 32, 34, 35, 37, 38, 40, 41, 43, 44] involved both digital and in-person support. For instance, Beleigoli et al (2020) included an online weight loss program and dietitian-delivered personalized feedback.

**Control and comparison.** A control or comparator group was present in 17 studies [22, 23, 25, 26, 28–34, 37, 38, 40, 41, 43]. There were a variety of control and comparison types adopted in the included studies. Three studies [25, 40, 43] adopted a waitlist control design where the control group did the same baseline assessment as the experimental group but were only involved in the same intervention upon the completion of the study. Five studies [26, 27, 32, 34, 37] reported no

intervention adopted in the control groups. Among the remaining 9 studies, one [30] received a self-monitored intervention, eight [22, 23, 28, 29, 31, 33, 38, 40] received the usual intervention without a digital component involved or a partial component. For example, Liu et al. (2015) only provided usual medical examinations to the control group without follow-up calls and text messages. Pillay et al. (2014) provided a partial intervention to the control group, sending general motivational email messages without personalised pedometer feedback.

**Physical, mental or other health-related measures.** All of the studies measured quantitative results, except for one study [42] that used qualitative and quantitative findings to develop an integrated technology-moderated institutional health promotion model. The outcome measures of the interventions were heterogeneous. Sixteen studies [22–26, 28, 30, 33, 34, 36, 38, 42, 44] assessed changes in physical health including PA, diet, and other lifestyle factors. Seven studies [27, 31, 32, 35, 37, 43, 45] measured changes in mental health factors, such as sleep, stress, work engagement and more. One study [29] measured improvements in occupational health and ergonomic conditions. Further details can be found in Table 3.

## Effectiveness of interventions

**Changes to weight.** Five studies reported weight as an outcome and four found that the WWPs had a significant effect on reducing weight. Among the four RCTs finding evidence of effectiveness, weight loss ranged from about 1–2 kilograms, including: web-assisted and telephone-assisted education (-1.08kg, p = 0.001; -1.92kg, p = 0.001) [28], web-based coach-delivered and computer-delivered feedback (-1.57 kg, [95%CI: –1.92 to –1.22], p = 0.001; –1.08kg, [95%CI: –1.41 to –0.75], p = 0.001) [41], phone messages and emails (-1.1kg [95%CI: -1.5 to -0.7], p<0.001) [34], and a pedometer and web-assisted exercise prescription (Males (BMI ≥24kg/m2) = -1.0kg [95%CI: –2.5 to –1.4], p<0.001; Females (BMI ≥24kg/m2) = -1.2kg [95% CI: –1.7 to –0.6], p<0.001]) [24]. One cohort study [23] of social media-based (WeChat) support and consultations found mixed evidence of effectiveness, with results suggesting the intervention was more effective among males and those who were actively engaged in the groups.

**Changes to physical activity level.** Four articles reported PA as a primary outcome, with three indicating a significant effect. Multiple intervention components were used in two studies. One prospective study [44] involved use of pedometers and a website for education and communication and found significant improvements in the intervention group (+3,519 steps/day; [95%CI: 3,484 to 3,553]; p <0.0001). Another quasi experimental study [26] using pedometers and a social media application (WeChat) for communication found a significant increase in walking from baseline to the end of intervention period (22%, p<0.05). There were two studies involving a single component. One RCT [25] that included online exercise videos led to a significant increase in PA (+5.8hrs/week, p = 0.04). One pilot study [40] that included a pedometer and motivational emails didn't not show a statistically significant improvement in steps between groups (IG = +996±1748 steps/day, CG = +97±750 steps/day).

**Changes to lifestyle beliefs, knowledge, behaviors, or chronic disease risk.** Seven studies reported lifestyle-related belief, knowledge, or behavioral outcomes or chronic disease risk as an outcome, with six indicating that WWP had a significant effect [22, 30, 32, 36, 39, 42]. For example, Liu, et al found a significant decrease in 10-year risk of CVD between the control and intervention groups in a cluster RCT ([95%CI: -4.47 to -1.18], *p* = 0.001) [22] and Ramachandran et al found a significantly lower incidence of type 2 diabetes ([95%CI: 0·45 to 0·92], p = 0·015) [41] in the intervention group. Only one study [38] did not find a statistically significant improvement in the primary outcome of interest (blood pressure) but did find a small reduction in body weight (−4·85kg, [95%CI: –8·21 to −1·48], p<0.05) and waist circumferences (−3·31cm, [95%CI: –5·95 to −0·67], p<0.05) in the intervention group of a parallel RCT.

**Changes to job performance.** Two studies reported job performance as outcome, with one RCT indicating that the WWP created a significant effect. The RCT [27] using a social media application (WeChat) for positive psychotherapy messages, showed significant improvement in job performance (job contribution: $F = 6.425$, $p = .013$; Task performance: $F = 29.252$, $p = .000$) and self-efficacy ($F = 13.326$, $p = .000$). One RCT [43] with 951 participants which involved an app-based cognitive behavioral therapy and messaging for technical support did not find a statistically significant effect at long term follow-up at 7 months (Program A: 95%CI: −0.11 to 0.12, p = 0.94; Program B: 95%CI:−0.08 to 0.16, p = 0.5).

**Changes to other health outcomes, such as stress, sleep and ergonomic condition.** Three studies reported stress as outcome and indicated that WWP was effective in reducing stress and burnout level among the employees. A RCT [35] conducted with 92 participants receiving online group yoga teaching showed significant reduction in stress, anxiety and depression level (p<0.001) immediately after the intervention, but not at 40 days (p = 0.49, p = 0.613, p = 0.563). A pilot study [37] conducted with 72 participants receiving online group educational treatment found evidence of reductions in stress, anxiety and depression level (95%CI:−5201 to −389, p<0.001; 95%CI: −35.18 to −29.16, p<0.001; 95%CI:−1.38 to—0.511, p<0.001). A pilot study [32] which assessed 81 participants adopting digital components of videos and email messages had shown significant improvement in quality of work life and resilience (r = 0.378, p<0.01; r = 0.365, p<0.05).

Two studies reported sleep quality as outcome with one of them finding a significant improvement to sleep quality post intervention. A pilot study [45] involving a sleep survey and recommendations delivered by tablet application showed significant effect on sleep awareness, total sleep duration during the weekend ($p = 0.046$), sleep debt ($p = 0.019$), sleep difficulties ($p<0.001$), and sleepiness ($p = 0.026$). Another RCT [31] using a multicomponent intervention involving both a messaging application (WhatsApp) for education and human support showed statistically significant improvement in other indicators, but not in overall sleep quality.

One quasi experimental study [29] involving a messaging application (Telegram), online education material, and text messages did not find significant improvement in workplace ergonomic condition.

## Feasibility and acceptability of interventions

Four studies assessed the feasibility and acceptability of the studies through questionnaires [25, 33, 34, 40] testing acceptability. All of them found that the interventions were feasible and/or acceptable. Blake et al. also identified facilitators to successful implementation, such as organizational support and participating in groups, as well as barriers, such as team leaders not adequately leading the exercise activities and limited space in the offices.

## Study quality

This scoping review did not include any formal quality assessment. Many of the studies were RCTs or quasi-experimental studies that aimed to reduce potential sources of bias. However, some of the limitations of the included studies were small sample sizes affecting the generalisability and results, and short study durations, leading to an unknown long-term effect of the intervention.

## Excluded or near-miss studies

Throughout the process of final study selection, a total of 22 studies were found to be highly relevant but did not meet certain inclusion criteria. Four of them were conducted in the wrong settings, as they did not take place in LMICs [47–50]. Articles were also excluded when the participants were born in LMICs but currently working in high-income countries, for

example, one article which was conducted among South Asian (India, Pakistan, or Bangladesh) workers in New York City was excluded. Three studies took place in atypical workplaces excluded from the current review (e.g., sex workers [46, 51], soldiers [52]). Four studies did not report health-related, work-related, or design-related outcomes. Four studies were excluded due to a lack of digital components in their workplace wellness interventions, including one article using a pedometer only for step count recording, but not to promote healthier behaviours [53].

During the screening stage, we found two articles from the same research study, but one was excluded because it was a protocol and therefore did not meet the inclusion criteria [43].

## Discussion

### Main findings

This scoping review aimed to assess the implementation of digital workplace wellness interventions in LMICs. Many studies were excluded from the review due to the fact that most digital workplace wellness research has been done in developed countries, such as the US and the UK. From this review, we see that digital workplace wellness interventions have been used to address a broad range of health behaviours (physical activity level, smoking cessation, sleep quality, burnout, etc.) in LMICs, but targeting outcomes with the goal of reducing the risk of chronic diseases seem to be most common. No other systematic or scoping reviews focus on the same criteria as our study, and this review therefore can help to further our understanding of digital workplace wellness interventions specifically in LMICs.

The final included 24 articles cover a wide range of interventions and measured outcomes. The content of the interventions varied, yet most of them involved mixed digital components, including websites, educational videos, social media or messaging applications, and phone calls or messages. As the studies showed a high level of heterogeneity in terms of intervention aims, digital components involved, outcomes and measures, it is difficult to compare or discern a clear pattern of effectiveness among the 24 studies.

Of the 24 articles, statistically significant improvements were found in all the studies except for two which found no changes [31, 38]. Therefore, it seems that digital WWPs hold promise for improving outcomes in LMICs, however, it is not possible to discern specific patterns between intervention components and outcomes as all the studies varied in intervention components, study design, duration, target outcomes and so on. For example, the duration of interventions varied greatly. Two studies that did not find statistically significant results had very different durations, one [31] was a 7-week intervention without follow up, and the other [38] was a 12 month intervention with 6 month follow up. The remaining 22 studies included interventions ranging from 2 weeks to 2 years (with or without follow-up) and found statistically significant improvements on the targeted outcomes, suggesting that there was no clear pattern of duration and follow-up for study effectiveness in the current review.

Generally, the studies concluded that digital interventions were well-accepted and feasible for the employees. Based on the findings of the reported studies, digital health interventions are potentially effective and feasible for improving employees' physical and mental wellbeing. The fact that digital interventions can be low-cost and more easily scalable have made them an attractive approach in low-resource settings. However, these findings are mixed and small in effect size, and long-term effects were not studied.

### Comparison with related literature

There are many studies on similar topics that did not fulfil all our inclusion criteria, particularly the criterion that the study should be conducted in an LMIC setting. Nonetheless, it is

worth discussing the currently available research on digital workplace wellness in high-income countries to compare our findings. For instance, a systematic review analysed the impact of pure digital health interventions in the workplace in high-income countries [54]. The review found that digital-only interventions can improve health-related outcomes in the workplace. They also found that they were more effective when tightly embedded in the work environment (such as downloading a software onto a work computer) and limited to distinct health behaviours that are regularly performed at work, such as physical activity and eating. On the other hand, more complex health behaviours that extend outside the workplace may require human support as a more effective approach [54]. Another systematic review also assessed mobile health interventions to encourage physical activity in the workplace in high-income countries [55]. It was found that commonly used behaviour change techniques were self-monitoring, feedback, goal-setting, and social comparison. Simultaneously, the main mHealth tools used were wearable activity trackers, smartphone apps, or both. Some studies also utilised text messaging, e-mails, social media groups or websites to deliver motivational messages. Approximately half of the studies found a significant increase in physical activity while 4 out of 10 studies reported significant reduction in sedentary time. The findings from these reviews in high-income settings are consistent with the findings of our review, which show that digital health workplace interventions can be feasible and effective [54–56]. Another scoping review [57] that examined the return on investment of WWPs in high-income countries found greater returns in larger companies (>500 employees), however, our scoping review was unable to identify any patterns by company size and many studies did not report this information.

Aside from digital health interventions in workplaces in LMICs, there are many studies examining digital interventions in LMICs, but without the focus on the workplace. One systematic review examined the use of short message service (SMS) interventions for disease prevention in developing countries [58]. The review concluded that, while there are many existing SMS applications for disease prevention, very limited evaluation is done to assess their effectiveness. It was also stated that the majority of the selected studies were from grey literature sources. Implementation barriers that were identified included language, timing of messages, network connection issues, high mobile phone turnover, data privacy and lack of financial incentives [58]. These same barriers might also apply to the implementation of digital workplace wellness interventions in LMICs, but the studies in this review did not discuss these barriers. Another literature review found 53 mHealth studies in LMICs [59]. However, the majority of these studies lacked a theoretical framework and outcome measures. Similar to our findings in workplaces specifically, these reviews suggest that there are small numbers of peer-reviewed studies that examine digital health in LMICs. In both cases, it is important to improve future work on digital health interventions in LMICs through the use of theoretical frameworks in the design of interventions and ensuring that programs are evaluated and measure health outcomes.

## Evidence gap

Based on our findings, there is a lack of information and evidence supporting the feasibility and effectiveness of digital health interventions among employees in LMICs. The locations of the studies only covered a few specific countries, mainly in China (n = 4), India (n = 4) and Iran (n = 4). Hence, future research is required as factors such as culture, ethnic groups, and lifestyle habits vary across countries and one successful intervention does not fit all.

## Future research directions

There are two implications for future research, based on the findings of this review. First, this review highlighted the relatively small amount of research that has been done on digital

workplace wellness interventions in LMICs, demonstrating the need for ongoing research in this area. Second, the review identified that there is no clear consensus on the theoretical frameworks that apply to the development of digital workplace wellness interventions in LMICs, which is likely due to the included studies involving varied approaches to health promotion, which may require different theoretical frameworks. These areas are also in need of future research to further our understanding of the mechanisms of workplace behaviour change and for which health outcomes digital workplace wellness is most effective.

## Strengths and limitations

The current review was conducted in accordance with PRISMA guidelines, including developing a robust search strategy, study selection, data extraction and synthesis. This review covered a broad range of study types involving quantitative and mixed method designs, the use of the most recent digital technologies in the studies, as well as targeting both physical and mental outcomes. One limitation is that the heterogeneous outcomes and incomplete reporting of the studies, affected the level of data synthesis that was possible. Another limitation was that non-peer-reviewed studies were not included, and relevant grey literature could have been missed.

## Conclusion

To our knowledge, this is the first scoping review to explore the nature of existing digital health interventions in workplace settings in LMICs. This scoping review gathered recent evidence of digital workplace wellness programs in LMICs. Based on our findings, there is relatively less evidence found when compared to developed or high-income countries. Positive improvements were found in the employees' mental and physical well-being with the implementation of digital health interventions, yet the effect of the interventions remain unclear in the LMIC context due to the small number of studies identified. Thus, there is a clear need for new high-quality studies with better reporting of interventions and outcomes to be conducted. Future studies should also adopt the use of theoretical frameworks into the research design, exploring more reliable and sustainable wellness programs to enhance the practice of digital health in the workplace in LMICs.

## Supporting information

**S1 Table. PRISMA checklist.**
(DOCX)

**S2 Table. Sample search strategy (MEDLINE).**
(DOCX)

**S3 Table. Detailed data extraction table.**
(DOCX)

## Author Contributions

**Conceptualization:** Tracy A. McCaffrey, Amutha Ramadas, Hema Malini, Jessica L. Watterson.

**Formal analysis:** Yi Chiann Thai, Deanna Sim.

**Investigation:** Yi Chiann Thai, Deanna Sim.

**Methodology:** Tracy A. McCaffrey, Amutha Ramadas, Hema Malini, Jessica L. Watterson.

**Project administration:** Jessica L. Watterson.

**Supervision:** Tracy A. McCaffrey, Amutha Ramadas, Jessica L. Watterson.

**Validation:** Yi Chiann Thai, Deanna Sim.

**Writing – original draft:** Yi Chiann Thai, Deanna Sim.

**Writing – review & editing:** Tracy A. McCaffrey, Amutha Ramadas, Hema Malini, Jessica L. Watterson.

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
