## [Decision Letter · Decision Letter 0]

17 Oct 2022

PONE-D-22-17661A Scoping Review of Digital Workplace Wellness Interventions in Low- and Middle-Income CountriesPLOS ONE

Dear Dr. Watterson,

Thank you for submitting your manuscript to PLOS ONE. After careful consideration, we feel that it has merit but does not fully meet PLOS ONE’s publication criteria as it currently stands. Therefore, we invite you to submit a revised version of the manuscript that addresses the points raised during the review process.

ACADEMIC EDITOR: As the reviewers have stated (see below) this is an important investigation in a needed topic area. However, the article is need of revisions in order to make it acceptable for publication. You have received two in-depth reviews that are clear and offer strong suggestions for ways to improve your paper. I agree with all of the points raised by the reviewers and hope that you take their comments into careful consideration if you decide to undertake a revision of the paper.

We look forward to receiving your revised manuscript.

Kind regards,

Ali A. Weinstein, Ph.D.

Academic Editor

PLOS ONE

Journal Requirements

2. We note that Figure 3 in your submission contain [map/satellite] images which may be copyrighted. All PLOS content is published under the Creative Commons Attribution License (CC BY 4.0), which means that the manuscript, images, and Supporting Information files will be freely available online, and any third party is permitted to access, download, copy, distribute, and use these materials in any way, even commercially, with proper attribution. For these reasons, we cannot publish previously copyrighted maps or satellite images created using proprietary data, such as Google software (Google Maps, Street View, and Earth). For more information, see our copyright guidelines: http://journals.plos.org/plosone/s/licenses-and-copyright.

a. You may seek permission from the original copyright holder of Figure 3 to publish the content specifically under the CC BY 4.0 license.  

Reviewers' comments:

Reviewer's Responses to Questions

**Comments to the Author**

1. Is the manuscript technically sound, and do the data support the conclusions?

Reviewer #1: Yes

Reviewer #2: Partly

2. Has the statistical analysis been performed appropriately and rigorously? 

Reviewer #1: Yes

Reviewer #2: N/A

3. Have the authors made all data underlying the findings in their manuscript fully available?

Reviewer #1: Yes

Reviewer #2: Yes

4. Is the manuscript presented in an intelligible fashion and written in standard English?

Reviewer #1: Yes

Reviewer #2: Yes

5. Review Comments to the Author

Reviewer #1: Overall, this article provides important and much-needed information on employer-sponsored digital workplace interventions in low and middle-income countries (LMICs). I applaud the authors for addressing this gap and conducting a thoughtful and rigorous scoping review of the literature. However, I have several major concerns regarding the Methods and Results. I believe addressing these concerns will make for a more informative, actionable, and relevant article.

MAJOR:

1) The stated goal of the paper was to provide a comprehensive synthesis of current evidence in relation to the effectiveness, feasibility, and acceptability of the digital workplace wellness intervention in the LMICs. However only very basic information is provided on the effectiveness, feasibility, and acceptability of programs. The Results section instead focuses on intervention and design components/qualities rather than findings related to their effectiveness, feasibility, and acceptability. Some of the latter elements are discussed in the third paragraph of the Discussion, but the main text Results do not adequately report these findings at a summary level. Information on effectiveness in the main article is largely in the form of a “conclusion” column in Table 2. I have two recommendations to address this concern:

-The Results section would benefit greatly from three high-level summary sections on effectiveness, feasibility, and acceptability (e.g., how well interventions worked, which health conditions or behaviors saw the most benefit, which intervention design types realized the most benefits, if information on effectiveness/feasibility/acceptability was mixed or lacking, etc.). I see very little reporting on the feasibility and acceptability of interventions. If it is absent in the underlying studies, please state this.

-S3 provides more informative results on effectiveness than Table 2. I recommend that the authors distill some of the information on statistical significance from S3 into a simplified column in Table 2, at least for the quantitative studies (which comprised all but one of the articles). Presenting the information in a simplified but study-specific column in Table 2 would help overcome the authors’ concern that it is difficult to report summary findings due to the difficulty in comparing or discerning “a clear pattern of effectiveness among the 24 studies.”

2) While the authors opine on the potential merits of digital-based interventions in LMICs, it seems that the limitations of such interventions in LMICs are overlooked. Please incorporate text into the Literature Review and Discussion sections that address this. For example, what are implications regarding generalizability of digital wellness interventions to certain areas of LMICs? Many areas of LMICs have poor access to broadband and electronic devices (e.g., home computers, internet, phone service, equipment cost, etc.). Additionally, I would recommend the authors add information about the size of the companies into their Results, and similarly, incorporate into the Literature Review any existing literature about the effectiveness of digital workplace interventions based on company size. One could imagine that tech-based wellness programs would be more cost-effective and successful for larger companies or those with greater fiscal infrastructure resources to implement the programs. The development and roll-out of digital interventions may require significant effort, finances, and resources that are not available to many employers in LMICs.

3) In general, the inclusion/exclusion criteria need to be more specific. Some of the information is presented in the Results, but the article would benefit from clarifying these inclusion/exclusion criteria in the Methods and in Table 1. For example, please address the following questions.

-It appears from the Results section (Study Characteristics) that pilot studies were included. Please clarify this in the Methods. This is important because the Table 1 exclusion criteria (“Protocols and study designs”) might imply that pilot studies were excluded, whereas the inclusion criteria of “design-related outcomes” and “feasibility” measures might imply that they were included.

-It appears from the Results section that interventions did not have to be employer-implemented (e.g., third party implementation such as that by an insurance company or government). Please include this information earlier on in the inclusion/exclusion criteria.

-Were there situations in which multiple distinct articles reported results from the same research study or wellness program and if so, how did you handle inclusion/exclusion of those articles?

-It appears from Table 2 that you included articles even if the wellness intervention was implemented for both LMICs and non-LMICs in the same sample. Please clarify this within Table 1 or somewhere in the main text, as well as whether results from the LMIC subsamples in those articles could be teased apart from the high-income countries, or in contrast, whether the pooled results were used.

-Students are listed as an excluded population in Table 1. However, Table 2 includes an article by Beleigoli (2020) that includes both students and university staff. Please clarify how this data was handled. Were you able to tease out data for staff from that of students?

MINOR:

4) I gather from S2 that the years searched were from 2010-2021. Please add this information to the main text in either lines 90-91 or somewhere nearby. Is there a specific rationale for choosing these years?

5) Line 103 Did each of the two reviewers screen all articles? Or were articles divided among the two reviewers? Please provide more detail on the allocation of articles to reviewers in in lines 118-119 as well.

6) In Table 2, please define all abbreviations (e.g., RCT, w/o) in the footnotes for the reader. Do not define abbreviations within the table (e.g., (MBSR, SE, SD, etc.). In the main text, there are instances when the same acronym is defined more than once (e.g., RCT in Study Characteristics and Participant Characteristics). Please carefully review the use of acronyms throughout the manuscript.

7) In the Results, please describe how many interventions were purely digital versus how many incorporated parallel in-person components.

8) In Figure 3, the color-coded legend has a max N of N=27,406. Should this not be N=27,466 to reflect the number of participants from India?

9) In the Participant Characteristics of the Results section, it would be helpful to describe whether studies targeted at-risk populations versus general employees overall.

10) Future Research Directions: Please elaborate on what is meant by the statement that there was “no clear consensus on applying theoretical frameworks to the development of digital workplace wellness interventions in LMICs”. I was actually surprised that more than half of the articles were theoretically grounded. In my experience, this proportion parallels what we see in employer-sponsored wellness interventions based in high-income countries.

11) Though no formal review was conducted on the quality of underlying studies/research articles, a sentence or two on the perceived quality of studies would facilitate interpretation of the findings.

Reviewer #2: This manuscript presents a scoping review that aims to explore and provide a comprehensive synthesis of current evidence in relation to the effectiveness, feasibility and acceptability of the digital workplace wellness intervention in the LMICs settings Low- and Middle Income Countries.

The authors utilize systematic searching methods that is guided by the Joanna Briggs Institute methodology and Preferred Reporting Items for Systematic reviews and Meta Analyses extension for Scoping Reviews (PRISMA-ScR) reporting guidelines. The review protocol was written prior to undertaking the review (OSF Registry: https://osf.io/qpr9j). The authors use various combinations of keywords related to “digital health”, “intervention”, “workplace”, and “developing country” in Ovid MEDLINE, EMBASE, CINAHL Plus, PsycINFO, Scopus and Cochrane Library for peer-reviewed articles in the English language. They identified 10,298 publications, of which 24 were included in the review based on their eligibility criteria, outlined in Table 1.

Their research questions are 1. How have digital technology interventions been conducted in the workplace in LMICs? 2. What research has been done, and what are the effects of these interventions on health- and job performance-related outcomes? 3. What research gaps can be identified from these interventions to improve health behaviours in the workplace in LMICs?

The authors present the PRISMA flow chart in Fig 1. The PRISMA flow chart and text clearly describe the methodology they used for identifying and selecting articles included in the review.

The authors performed narrative data synthesis according to 6 groups of identified outcomes: Lifestyle/ Chronic disease Risk (A), including smoking and cardiovascular disease risk, Weight Management (B), Physical Activity (C), Job performance (D) including work engagement and ergonomic conditions, Stress (E) including burnout, depression and anxiety, and Sleep (F). However, there is little discussion in the findings section about the types of interventions used and the focus or goals of the interventions. What were the goals/focus of the interventions included in the systematic review? Providing more information about the interventions would improve the transferability of the findings to future research and for practice.

Little information is discussed on the duration of intervention and follow-up, additional explanation on this could help the reader understand whether duration and follow-up were critical to effectiveness of the studies.

The authors provide a great discussion of theoretical frameworks used in the study. This information could be more understandable if connected to the types of interventions delivered.

Be specific on the number of studies with a specific attribute or methodology. For example, “Some studies utilized multicomponent interventions” could be improved by including the exact number of studies, similar to reporting earlier “Seven studies included interventions led by both the research team and participants.” (page 20)

More explanation is needed to describe the types of control and comparison groups. (Page 21, line 239.)

If space is needed, consider removing the section titled Excluded or near-miss studies or moving this section to an appendix. (line 250 – 258)

The authors provide in-depth information in table format. Table 2. summarizes included studies grouped by main targeted health outcome. Information presented includes Study, Country, Study design Participants Study duration Mode of delivery / Underlying theory Measured Outcomes Conclusion. Another table provides information on study characteristics such as type of research, workplace settings, study location and publication year; theoretical frameworks; participant characteristics such as sample size and gender. The authors provide supplementary materials on more details on the studies included in S4 Table. These tables provide a lot of detailed information on the studies. The authors may consider condensing some of this information to help the reader quickly understand the results of the studies, perhaps by including a table that lists the number of studies including a specific outcome and whether the outcome was positive significant, negative significant, non-significant, or not reported.

The discussion section states, “This review shows that digital workplace wellness interventions have been used to address a broad range of health behaviours (physical activity level, smoking cessation, sleep quality, burnout) in LMICs,” but there is no explanation in the findings section on the types of interventions or the outcomes associated with those interventions. (page 22)

The discussion section also states that “It is not easy to compare or discern a clear pattern of effectiveness among the 24 studies,” (page 22) while the abstract includes the following statement that indicates positive findings for effectiveness “Most of the studies reported positive feedback on the use of digital wellness interventions in workplace settings. Modest evidence suggests that digital workplace wellness interventions were feasible, cost-effective and acceptable.” (page 2) and “Positive findings and significant improvements were found in all the studies except for two, which found no changes” (page 22). The authors may consider updating the language to remove confusion around effectiveness and positive findings and to address the question: Why was it hard to compare or discern a pattern of effectiveness?

There is a great discussion on evidence gaps that identifies key points to consider for future research.

The authors conclude that their review identified “no clear consensus on applying theoretical frameworks to the development of digital workplace wellness interventions in LMICs, nor on the outcomes that should be targeted or evaluated.” The lack of consensus on theoretical frameworks may be due to included studies involved varying approaches to health promotion and focus on varying outcome measures, which require different theoretical frameworks.

6. PLOS authors have the option to publish the peer review history of their article (what does this mean?). If published, this will include your full peer review and any attached files.

Reviewer #1: **Yes: **Mary Louise Pomeroy

Reviewer #2: **Yes: **Debora Goetz Goldberg

---

## [Author Response · Author response to Decision Letter 0]

9 Jan 2023

REVIEWER COMMENTS 

Reviewer #1 

Overall, this article provides important and much-needed information on employer-sponsored digital workplace interventions in low and middle-income countries (LMICs). I applaud the authors for addressing this gap and conducting a thoughtful and rigorous scoping review of the literature. However, I have several major concerns regarding the Methods and Results. I believe addressing these concerns will make for a more informative, actionable, and relevant article.

Major:

1) The stated goal of the paper was to provide a comprehensive synthesis of current evidence in relation to the effectiveness, feasibility, and acceptability of the digital workplace wellness intervention in the LMICs. However only very basic information is provided on the effectiveness, feasibility, and acceptability of programs. The Results section instead focuses on intervention and design components/qualities rather than findings related to their effectiveness, feasibility, and acceptability. Some of the latter elements are discussed in the third paragraph of the Discussion, but the main text Results do not adequately report these findings at a summary level. Information on effectiveness in the main article is largely in the form of a “conclusion” column in Table 2. I have two recommendations to address this concern:

-The Results section would benefit greatly from three high-level summary sections on effectiveness, feasibility, and acceptability (e.g., how well interventions worked, which health conditions or behaviors saw the most benefit, which intervention design types realized the most benefits, if information on effectiveness/feasibility/acceptability was mixed or lacking, etc.). I see very little reporting on the feasibility and acceptability of interventions. If it is absent in the underlying studies, please state this.

Response to 1.1: Thank you for this suggestion. We have added two sections for “Effectiveness” and “Feasibility and Acceptability” in the Results section. Feasibility and acceptability were combined into one section due to the small number of studies reporting any outcomes related to these topics (as you stated). 

-S3 provides more informative results on effectiveness than Table 2. I recommend that the authors distill some of the information on statistical significance from S3 into a simplified column in Table 2, at least for the quantitative studies (which comprised all but one of the articles). Presenting the information in a simplified but study-specific column in Table 2 would help overcome the authors’ concern that it is difficult to report summary findings due to the difficulty in comparing or discerning “a clear pattern of effectiveness among the 24 studies.”

Response to 1.2: Thank you for the suggestion. We have updated Table 2 with more details.

2) While the authors opine on the potential merits of digital-based interventions in LMICs, it seems that the limitations of such interventions in LMICs are overlooked. Please incorporate text into the Literature Review and Discussion sections that address this. For example, what are the implications regarding generalizability of digital wellness interventions to certain areas of LMICs? Many areas of LMICs have poor access to broadband and electronic devices (e.g., home computers, internet, phone service, equipment cost, etc.). Additionally, I would recommend the authors add information about the size of the companies into their Results, and similarly, incorporate into the Literature Review any existing literature about the effectiveness of digital workplace interventions based on company size. One could imagine that tech-based wellness programs would be more cost-effective and successful for larger companies or those with greater fiscal infrastructure resources to implement the programs. The development and roll-out of digital interventions may require significant effort, finances, and resources that are not available to many employers in LMICs.

Response to 2: Thank you for these suggestions. When available, the company sizes were added in Table 2. Most studies did not report the company sizes, so the number of participants included for baseline eligibility screening was reported instead. The following was also added to the Literature Review regarding company size: 

“Another scoping review that examined the return on investment of WWPs in high-income countries found greater returns in larger companies (>500 employees), however, our scoping review was unable to identify any patterns by company size and many studies did not report this information.”

An additional sentence was added to the Discussion section to expand upon the LMIC barriers that were identified in other mHealth studies (and those that you mention here):

“Implementation barriers that were identified included language, the timing of messages, network connection issues, high mobile phone turnover, data privacy, and lack of financial incentives. These same barriers might also apply to the implementation of digital workplace wellness interventions in LMICs, but the studies in this review did not discuss these barriers.”

3) In general, the inclusion/exclusion criteria need to be more specific. Some of the information is presented in the Results, but the article would benefit from clarifying these inclusion/exclusion criteria in the Methods and in Table 1. For example, please address the following questions.

-It appears from the Results section (Study Characteristics) that pilot studies were included. Please clarify this in the Methods. This is important because the Table 1 exclusion criteria (“Protocols and study designs”) might imply that pilot studies were excluded, whereas the inclusion criteria of “design-related outcomes” and “feasibility” measures might imply that they were included.

Response to 3.1: Thanks for the comment. We have updated the study design inclusion criterion in Table 1 under the “Methods” Section.

-It appears from the Results section that interventions did not have to be employer-implemented (e.g., third party implementation such as that by an insurance company or government). Please include this information earlier on in the inclusion/exclusion criteria.

Response to 3.2: Thanks for the comment. We have added an implementer inclusion criterion in Table 1 under the “Methods” Section.

-Were there situations in which multiple distinct articles reported results from the same research study or wellness program and if so, how did you handle inclusion/exclusion of those articles?

Response to 3.3: Yes, there was one set of articles from the same research study, but one of the two articles was the study protocol and was excluded accordingly. The second article (with the study results) was included. We have added this detail under the “Excluded or near-miss studies” section:

“During the screening stage, we found two articles from the same research study but one was excluded because it was a protocol and therefore did not meet the inclusion criteria.”

-It appears from Table 2 that you included articles even if the wellness intervention was implemented for both LMICs and non-LMICs in the same sample. Please clarify this within Table 1 or somewhere in the main text, as well as whether results from the LMIC subsamples in those articles could be teased apart from the high-income countries, or in contrast, whether the pooled results were used.

Response to 3.4: Thanks for the comment. We have updated the location inclusion criterion in Table 1 under the “Methods” Section, and also added the following sentence in the “Study selection and characteristics” section:

“For these studies including participants from both LMICs and high-income countries, results could not be separated by country so the pooled results are presented in this review.”

-Students are listed as an excluded population in Table 1. However, Table 2 includes an article by Beleigoli (2020) that includes both students and university staff. Please clarify how this data was handled. Were you able to tease out data for staff from that of students?

Response to 3.5: Thanks for the comment. We have updated the target population inclusion criterion in Table 1 under the “Methods” Section, and also added the following sentence in the “Participant characteristics” section:

“The study by Beleigoli et al. included both staff and students of a university and, though studies of students did not meet the inclusion criteria for this review, the pooled results are presented here as it was not possible to separate the results for staff only.”

Minor:

4) I gather from S2 that the years searched were from 2010-2021. Please add this information to the main text in either lines 90-91 or somewhere nearby. Is there a specific rationale for choosing these years?

Response to 4: Thanks for this question. The explanation has been added under “Data sources and search strategy” section:

“Articles were only included from 2010-2021 to avoid the inclusion of obsolete digital components such as CD-ROMs and personal digital assistants (PDAs) which are not applicable in the current digital era.” 

5) Line 103 Did each of the two reviewers screen all articles? Or were articles divided among the two reviewers? Please provide more detail on the allocation of articles to reviewers in lines 118-119 as well.

Response to 5: Thanks for the question. We have added details to further clarify the allocation of articles and tasks distributed under the “Study Selection” and “Data Extraction” Sections: 

Study Selection

“Both reviewers screened all the articles.”

Data extraction

“Two reviewers conducted the data extraction from half of the 24 finalised articles (12 each).”

6) In Table 2, please define all abbreviations (e.g., RCT, w/o) in the footnotes for the reader. Do not define abbreviations within the table (e.g., (MBSR, SE, SD, etc.). In the main text, there are instances when the same acronym is defined more than once (e.g., RCT in Study Characteristics and Participant Characteristics). Please carefully review the use of acronyms throughout the manuscript.

Response to 6: Thanks for the comment - we've reviewed and updated them.

7) In the Results, please describe how many interventions were purely digital versus how many incorporated parallel in-person components. (1 sentence)

Response to 8: Thanks for raising this point - further details have been added under the “Digital Component” section:

Digital component

“... Of the 24 studies, 7 studies24,25,33,36,39,42,45 involved pure digital intervention while the remaining 16 studies 22,23,26,28-32,34,35,37,38,40,41,43,44 involved both digital and human support. For instance, Beleigoli et al (2020) included an online weight loss program and dietitian-delivered personalized feedback.”

8) In Figure 3, the color-coded legend has a max N of N=27,406. Should this not be N=27,466 to reflect the number of participants from India?

Response to 8: Thanks for catching this - the figure has been updated with the correct number.

9) In the Participant Characteristics of the Results section, it would be helpful to describe whether studies targeted at-risk populations versus general employees overall.

Response to 9: Thanks for this suggestion. We have added a “Target population” subsection under the “Results” section to further clarify this:

“17 studies targeted general employees without any health condition and seven targeted the at-risk population. Four articles employed a weight management intervention with an overweight or obese population (n=3) or population intending to lose weight (n=1). One article included employees having prehypertension, one included employees who showed high-stress symptoms, and one included employees with a family history of risk factors for metabolic diseases. Of the 24 articles, two targeted male-only and female-only populations, the remaining included both genders.”

10) Future Research Directions: Please elaborate on what is meant by the statement that there was “no clear consensus on applying theoretical frameworks to the development of digital workplace wellness interventions in LMICs”. I was actually surprised that more than half of the articles were theoretically grounded. In my experience, this proportion parallels what we see in employer-sponsored wellness interventions based in high-income countries.

Response to 10: Thanks for raising this point. We have added further details to clarify that we mean a range of different theoretical frameworks are being studied:

“Second, the review identified that there is no clear consensus on the theoretical frameworks that apply to the development of digital workplace wellness interventions in LMICs, which is likely due to the included studies involving varied approaches to health promotion, which may require different theoretical frameworks.” 

11) Though no formal review was conducted on the quality of underlying studies/research articles, a sentence or two on the perceived quality of studies would facilitate interpretation of the findings.

Response to 11: Thanks for this suggestion. We have added a “Study Quality” section to further clarify this: 

“This scoping review did not include any formal quality assessment. Many of the studies were RCTs or quasi-experimental studies that aimed to reduce potential sources of bias. However, some of the limitations of the included studies were small sample sizes affecting the generalisability and results, and short study durations, leading to an unknown long-term effect of the intervention.”

Reviewer #2: This manuscript presents a scoping review that aims to explore and provide a comprehensive synthesis of current evidence in relation to the effectiveness, feasibility and acceptability of the digital workplace wellness intervention in the LMICs settings Low- and Middle Income Countries.

The authors utilize systematic searching methods that is guided by the Joanna Briggs Institute methodology and Preferred Reporting Items for Systematic reviews and Meta Analyses extension for Scoping Reviews (PRISMA-ScR) reporting guidelines. The review protocol was written prior to undertaking the review (OSF Registry: https://osf.io/qpr9j). The authors use various combinations of keywords related to “digital health”, “intervention”, “workplace”, and “developing country” in Ovid MEDLINE, EMBASE, CINAHL Plus, PsycINFO, Scopus and Cochrane Library for peer-reviewed articles in the English language. They identified 10,298 publications, of which 24 were included in the review based on their eligibility criteria, outlined in Table 1.

Their research questions are 1. How have digital technology interventions been conducted in the workplace in LMICs? 2. What research has been done, and what are the effects of these interventions on health- and job performance-related outcomes? 3. What research gaps can be identified from these interventions to improve health behaviours in the workplace in LMICs?

The authors present the PRISMA flow chart in Fig 1. The PRISMA flow chart and text clearly describe the methodology they used for identifying and selecting articles included in the review.

The authors performed narrative data synthesis according to 6 groups of identified outcomes: Lifestyle/ Chronic disease Risk (A), including smoking and cardiovascular disease risk, Weight Management (B), Physical Activity (C), Job performance (D) including work engagement and ergonomic conditions, Stress (E) including burnout, depression and anxiety, and Sleep (F). However, there is little discussion in the findings section about the types of interventions used and the focus or goals of the interventions. What were the goals/focus of the interventions included in the systematic review? Providing more information about the interventions would improve the transferability of the findings to future research and for practice.

Response to 1: Thanks for this suggestion. The goal/focus of the intervention was added in Table 2 under the “intervention details” column.

Little information is discussed on the duration of intervention and follow-up, additional explanation on this could help the reader understand whether duration and follow-up were critical to effectiveness of the studies.

Response to 2: Thanks for raising this point. We have added some details regarding the study durations and follow-up periods in the “Main Findings” subsection of the “Discussion” section:

“For example, the duration of interventions varied greatly. Two studies that did not find statistically significant results had very different durations - one was a 7-week intervention without follow up, and the other was a 12-month intervention with 6 month follow up. The remaining 22 studies included interventions ranging from 20 minutes to 2 years (with or without follow-up) and found statistically significant improvements on the targeted outcomes, suggesting that there was no clear pattern of duration and follow-up for study effectiveness in the current review.”

The authors provide a great discussion of theoretical frameworks used in the study. This information could be more understandable if connected to the types of interventions delivered.

Response to 3: Thank you for the suggestion - we’ve added more details under the “Theoretical Frameworks” section:

“Some of the included studies drew on theoretical frameworks in their design or analysis. Three studies reported that their interventions drew on the Transtheoretical Model of Behaviour Change (n=3) with two using motivational messages and calls, and the other providing health information. Two studies were based on Social Cognitive Theory (n=2), with one involving education/training, and the other using a WeChat group for motivation and progress reporting. Two used the Health Belief Model (n=2), with one involving education material and the other involving motivational calls and personal text messages. One study adopted the Theory of Planned Behaviour with education training, messaging, and knowledge sharing in a Telegram group. Another used Behaviour Change Techniques with coaching and pedometer-generated personalised feedback. Another used Self Efficacy while asking participants to post 3 good things every day. One study used Goal Setting and provided health information through phone messages and emails. Another study drew on the Behaviour Change Wheel and involved coaching and pedometer-generated personalised feedback. Another study used Influential Theory and involved pictures, videos, and text messages on positive emotions. Eleven studies mentioned no clear theoretical basis.”

Be specific on the number of studies with a specific attribute or methodology. For example, “Some studies utilized multicomponent interventions” could be improved by including the exact number of studies, similar to reporting earlier “Seven studies included interventions led by both the research team and participants.” (page 20)

Response to 4: Thanks for raising this point - we've updated the exact number of studies: 

“10 studies23,32,33,35-39,43,45 involved single digital component intervention, whereas 14 studies22,24-31,34,40,41,43,44 utilised multi-component interventions where more than one digital component was used.”

More explanation is needed to describe the types of control and comparison groups. (Page 21, line 239.) If space is needed, consider removing the section titled Excluded or near-miss studies or moving this section to an appendix. (line 250 – 258)

Response to 5: Thanks for this suggestion. Further details have been added under the “Control and Comparison” section:

“Five studies reported no intervention adopted in the control groups. Among the remaining 11 studies, one received a self-monitored intervention, eight received the usual intervention without a digital component involved or a partial component. For example, Liu et al. (2015) only provided usual medical examinations to the control group without follow-up calls and text messages. Pillay et al. (2014) provided a partial intervention to the control group, sending general motivational email messages without personalised pedometer feedback.”

The authors provide in-depth information in table format. Table 2. summarizes included studies grouped by main targeted health outcome. Information presented includes Study, Country, Study design Participants Study duration Mode of delivery / Underlying theory Measured Outcomes Conclusion. Another table provides information on study characteristics such as type of research, workplace settings, study location and publication year; theoretical frameworks; participant characteristics such as sample size and gender. The authors provide supplementary materials on more details on the studies included in S4 Table. These tables provide a lot of detailed information on the studies. The authors may consider condensing some of this information to help the reader quickly understand the results of the studies, perhaps by including a table that lists the number of studies including a specific outcome and whether the outcome was positive significant, negative significant, non-significant, or not reported.

Response to 6: Thank you for this suggestion. A new Table 1 has been added with the overall characteristics of the included studies, and further outcome details have been added to Table 2. New subsections on “Effectiveness of the Interventions” and “Feasibility and Acceptability of the Interventions” have also been added to the Results to help readers quickly understand the results of the studies. 

The discussion section states, “This review shows that digital workplace wellness interventions have been used to address a broad range of health behaviours (physical activity level, smoking cessation, sleep quality, burnout) in LMICs,” but there is no explanation in the findings section on the types of interventions or the outcomes associated with those interventions. (page 22)

Response to 7: Thanks for raising this point. We have added a new subsection on the “Effectiveness of the Interventions” in the Results section which further clarifies the types of intervention and the associated outcomes.

The discussion section also states that “It is not easy to compare or discern a clear pattern of effectiveness among the 24 studies,” (page 22) while the abstract includes the following statement that indicates positive findings for effectiveness “Most of the studies reported positive feedback on the use of digital wellness interventions in workplace settings. Modest evidence suggests that digital workplace wellness interventions were feasible, cost-effective and acceptable.” (page 2) and “Positive findings and significant improvements were found in all the studies except for two, which found no changes” (page 22). The authors may consider updating the language to remove confusion around effectiveness and positive findings and to address the question: Why was it hard to compare or discern a pattern of effectiveness?

Response to 8: Thanks for raising this point. We have updated the Results section to more clearly articulate the results related to effectiveness, feasibility, and acceptability (as described in our responses to your earlier comments above), and have added details to address your question in the “Main findings” subsection of the Discussion section:

“Of the 24 articles, statistically-significant improvements were found in all the studies except for two which found no changes. Therefore, it seems that digital WWPs hold promise for improving outcomes in LMICs, however, it is not possible to discern specific patterns between intervention components and outcomes as all the studies varied in intervention components, study design, duration, target outcomes, and so on.

There is a great discussion on evidence gaps that identifies key points to consider for future research. The authors conclude that their review identified “no clear consensus on applying theoretical frameworks to the development of digital workplace wellness interventions in LMICs, nor on the outcomes that should be targeted or evaluated.” The lack of consensus on theoretical frameworks may be due to included studies involved varying approaches to health promotion and focus on varying outcome measures, which require different theoretical frameworks.

Response to 9: Thank you for raising this point. We have added this potential explanation to the “Future Research Directions” subsection of the Discussion section:

“Second, the review identified that there is no clear consensus on the theoretical frameworks that apply to the development of digital workplace wellness interventions in LMICs, which is likely due to the included studies involving varied approaches to health promotion, which may require different theoretical frameworks.”

---

## [Decision Letter · Decision Letter 1]

8 Feb 2023

A Scoping Review of Digital Workplace Wellness Interventions in Low- and Middle-Income Countries

PONE-D-22-17661R1

Dear Dr. Watterson,

We’re pleased to inform you that your manuscript has been judged scientifically suitable for publication and will be formally accepted for publication once it meets all outstanding technical requirements.

Kind regards,

Ali A. Weinstein, Ph.D.

Academic Editor

PLOS ONE

Additional Editor Comments (optional):

Thank you for thoughtfully addressing the reviewers' comments.

Reviewers' comments:

Reviewer's Responses to Questions

**Comments to the Author**

1. If the authors have adequately addressed your comments raised in a previous round of review and you feel that this manuscript is now acceptable for publication, you may indicate that here to bypass the “Comments to the Author” section, enter your conflict of interest statement in the “Confidential to Editor” section, and submit your "Accept" recommendation.

Reviewer #1: All comments have been addressed

Reviewer #2: All comments have been addressed

2. Is the manuscript technically sound, and do the data support the conclusions?

Reviewer #1: Yes

Reviewer #2: Yes

3. Has the statistical analysis been performed appropriately and rigorously? 

Reviewer #1: Yes

Reviewer #2: N/A

4. Have the authors made all data underlying the findings in their manuscript fully available?

Reviewer #1: Yes

Reviewer #2: Yes

5. Is the manuscript presented in an intelligible fashion and written in standard English?

Reviewer #1: Yes

Reviewer #2: Yes

6. Review Comments to the Author

Reviewer #1: (No Response)

Reviewer #2: The authors were very comprehensive in their edits to the manuscript based on feedback from multiple reviewers.

7. PLOS authors have the option to publish the peer review history of their article (what does this mean?). If published, this will include your full peer review and any attached files.

Reviewer #1: No

Reviewer #2: **Yes: **Debora Goetz Goldberg

---

## [Editor Report · Acceptance letter]

20 Feb 2023

PONE-D-22-17661R1 

A Scoping Review of Digital Workplace Wellness Interventions in Low- and Middle-Income Countries 

Dear Dr. Watterson:

I'm pleased to inform you that your manuscript has been deemed suitable for publication in PLOS ONE. Congratulations! Your manuscript is now with our production department. 

Kind regards, 

on behalf of

Dr. Ali A. Weinstein 

Academic Editor

PLOS ONE